# Evaluation of the probable annual flood damage influenced by El-Niño in the Kan River Basin, Iran

Farhad Hooshyaripor[1], Sanaz Faraji-Ashkavar[2], Farshad Koohyian[3], Qiuhong Tang[4,5*], Roohollah Noori[6]

[1]Department of Civil Engineering, Architecture and Art, Science and Research Branch, Islamic Azad University, Tehran, 1477893855, Iran
[2]Department of Civil Engineering, Al-Taha University, Tehran, 1488836164, Iran
[3]Water Research Institute, Ministry of Energy, Tehran, 1658954381, Iran
[4]Key Laboratory of Water Cycle and Related Land Surface Processes, Institute of Geographic Sciences and Natural Resources Research, Chinese Academy of Sciences, Beijing, 100101, China
[5]University of Chinese Academy of Sciences, Beijing, 100049, China
[6]School of Environment, College of Engineering, University of Tehran, Tehran, 141785311, Iran

*Correspondence to*: Qiuhong Tang (tangqh@igsnrr.ac.cn)

**Abstract.** Although many studies have explored the effect of teleconnection patterns on flood, few investigations have focused on the assessment of expected flood damages under such large-scale atmospheric signals. This study aims at determining the effect of the most emblematic teleconnection, El-Niño, on the expected damages of floods with low return periods in Kan River basin, Iran. To determine the flood damage costs, the annual precipitation enhancement during El-Niño condition was firstly estimated using a probabilistic approach and the inundation area was then determined under high probability levels of increased rainfall due to El-Niño for 5-, 10- and 50-year return period floods. The results showed that El-Niño increases the precipitation amount up to 8.2% and 31% with 60% and 90% confidence level, respectively. Flood damage assessment using damage-elevation curves showed that the expected increase percentile in flood damage for smaller return periods, which is more frequent, is much more than that for larger return periods. In general, for the return periods of 5- and 10- year, 31% increase in the precipitation would result in 2416% and 239% damage increase, respectively. However, for the 50-year rainfall this increase amount will be about 74%. These results indicate the importance of small flood events in flood management planning during El-Niño.

## 1 Introduction

In recent decades, the frequency of flood events and the resultant damages have been increasing dramatically in Iran. Reviewing the recorded flood events in the past decades shows an increasing trend. According to the available reports, the number of flooding events happened in any parts of Tehran over four decades had grown up from 12 cases in 1951 to 54 cases in 1991 (Farsnews 2015; Saghafian et al. 2017). Climate change, approaching to floodplain, land use changes, diversion of the waterways, population increase and destructive effects of human activities, degradation of forests and pastures, construction

of dysfunctional and vulnerable hydraulic structures can be mentioned as the reasons for increasing flood risks (Wang et al. 2019; Tang 2020).

The magnitude and frequency of flood events in each region depends on several factors: (i) physiographical features of the catchment such as shape, slope, and rivers network density, (ii) hydrological features such as precipitation, storage and initial losses, evapotranspiration, and permeability, (iii) human activities, (iv) large-scale atmospheric signals and (v) climate change. These factors affect the occurrence frequency and intensity of the flood and consequently the amount of damage costs. Identifying these factors will help to manage the flood and reduce the risks. In recent years, the effects of teleconnection phenomena have been more and more discussed and has tried to identify their impact on the local climate (Nazemosadat and Ghasemi 2004; Ward et al. 2014; Saghafian et al. 2017; Alizadeh-Choobari et al. 2018; Hooshyaripor et al. 2018; Hao et al. 2018).

Prediction of teleconnection indicators helps to reduce the flood damages by implementing the necessary practical measures (Schöngart and Junk 2007). Sun et al. (2015) showed that parts of North and South America, South and East Asia, South Africa, Australia and Europe are affected by El-Niño Southern Oscillation (ENSO). Grieco and DeGaetano (2018) concluded that the occurrence of El-Niño in the winter reduces the frequency of high waves in the east of the Ontario Lake, while there was no meaningful relationship in the conditions of La-Niña. Azmoodehfar and Azarmsa (2013) showed that the south-east of Iran during the years with the event of La-Niña experienced a higher-than-normal maximum and minimum temperature. Schöngart and Junk (2007) showed that there is a strong correlation between ENSO and Amazon River flood such that the river water level decreases in the warm episode of the ENSO (El-Niño) and increases in its cold episode (La-Niña). In a global study, Ward et al. (2014) showed the impact of ENSO on the daily peak discharge of some important rivers.

In addition, many studies have shown the effect of ENSO on the climate variability in Iran. Nazemosadat and Ghasemi (2004) indicated that El-Niño is associated with wet periods over most regions of Iran during autumn and winter while the risk of droughts is high during La Niña. Their study revealed that El-Niño has the least influence over the southeastern and northwestern regions of the country during winter. Haghnegahdar et al. (2007) showed that in the Dez and Karoun basins (located in the west and southwest of Iran) during March and April the El-Niño increases the probability of occurrence of maximum annual flood and vice versa in the case of La-Niña. Their results showed that the maximum annual flood variation during El-Niño events is much greater than that during La-Niña events. Gholizadeh (2015) investigated the effect of El-Niño on the rainfall events between 1973 and 2012 and concluded that the El-Niño phenomenon increases the annual rainfall of Iran while its severity reduces from south to north of country. Hooshyaripor et al. (2019) investigated the impact of different teleconnection indices on Iran rainfall, and concluded that El-Niño can enhance the annual precipitation by nearly 40%. They stated that due to the increase of rainfall, the river discharge will be affected directly. So far, many studies have focused on the impact of El-Niño over river flow. Alizadeh-Choobari et al. (2018) indicated that the ENSO cycle contributes to the interannual climate variability over Iran. According to their results, about 26% of the variance in annual precipitation over Iran is related to the El-Niño. Based on their achievements, In spite of the seasonality of the ENSO signal and its interevent variability,Iran

is anomalously wet during the EP El-Niño and dry during La Niña and the impacts of La Niña and the EP El-Niño are generally stronger over the warm and arid regions of Iran.

Although, the effect of ENSO on the precipitation has been frequently studied in Iran, there are few studies about ENSO influence on the socioeconomic impacts of floods even around the world (Ward et al. 2014). The main reason for the limited research on the economic impacts of climate and hydrologic variability is said to be the lack of economic data on flood damages (Changnon 2003). Analyzing the National Flood Insurance Program daily claims and losses and Multivariate ENSO Index (MEI), Corringham and Cayan (2019) quantified insured flood losses across the western United States from 1978 to 2017. They showed that in coastal Southern California and across the Southwest of the United States, El-Niño has had a strong effect in producing more frequent and higher magnitudes of insured losses, while in the Pacific Northwest, the opposite pattern with weaker and less spatially coherent has been reported. Changnon (2003) revealed that the strong El-Niño events of 1982/83 and 1997/98 have caused significant flood damages over $2.8 billion in Southern California. Null (2014) demonstrated that from 1949 until 1997 out of the six seasons that flood damages costs exceeded $1 billion in California three cases had been El-Niño years; one very strong (1982), one moderate (1994) and one weak (1968). Ward et al. (2014) showed that ENSO exerts strong and widespread influences on both flood hazard and risk. They assessed ENSO's influence in terms of affected population, gross domestic product and economic damages on the flood risk at the global scale and showed that climate variability, especially from ENSO, should be incorporated into disaster-risk analyses and policies. They revealed that, if the frequency and/or magnitude of ENSO events were to change in the future due to climate change, change in flood-risk variations across almost half of the world's terrestrial regions is happened. Ward et al. (2016) provided a global modelling exercise to examine the relationships between flood duration and frequency and ENSO. They indicated that the duration of flooding compared to flood frequency is more sensitive to ENSO.

These studies indicate the importance of teleconnection in the flood characteristics in many parts of the world. To the best of the authors' knowledge few investigations have focused on the assessment of the expected damages under the El-Niño or La-Niña condition. Obviously, damage assessment is an important part of the flood risk analysis, which determines the need for flood management programs and their priorities. The question addressed in this research is that, given the increasing impact of rainfall due to El-Niño, how much losses/damages are expected to be added in a specified study area. To answer this question, the catchment of Kan River in north of Tehran metropolis was selected. Due to the importance of the Basin, numerous flood risk studies have been conducted in this area. Yazdi and Salehi Neyshabouri (2012) coupled the MIKE-11 hydrodynamic model and the NSGA-II multi-objective optimization model to find the optimal combination of structural and non-structural methods for flood risk management. Their purpose was to reduce the expected flood damages in which probabilistic damage curves were used to assess the impact of different methods on flood mitigation. In another study, Yazdi et al. (2013) evaluated the effect of non-structural watershed management -one of the proposed flood management plans- on the possible damages. Hooshyaripor et al. (2015) considered social factors to find the optimal combination of flood management methods with minimum investment and the highest potential for damage reduction. They provided a methodology for reducing social

vulnerability in flood management planning. Hence, this paper focuses on the impact of El-Niño not only on the precipitation amount but also on the flood damages that are expected to be increased due to the El-Niño event.

## 2 Study area

Kan River basin is one of the most important flooding basins located in the north of Tehran city and a vulnerable area against
flood (WRI 2011a; Yazdi and Salehi Neyshabouri 2012; Hooshyaripor et al. 2015). It is reported that in the flood of July 15, 2015 a 20-minute storm caused 8 losses of life, several bridge and diversion dam failures, and more than 10 million dollars in damages to the residential, commercial and agricultural areas (ISNA 2015). In June 1968 a heavy rain as 6 times as the annual precipitation happened in 2 days had caused 31 losses of life in Tehran central area and huge damage to the properties. In general, during a period of 60-year (from 1954 to 2015) at least 8 flood events that resulted in loss of life (in total 2200 people)
have been reported in Kan and central Tehran areas. Existence of many restaurants, demographic, recreational, tourist and pilgrimage centers adjacent to the sloping Kan River have exacerbated the potential for damage (WRI 2011a).

The Kan basin is located between the longitude of the 51º 10' and the 51º 23' and the latitude of 35º 45' to 35º 58'. The basin can be divided into 10 sub-basins (Figure 1**Error! Reference source not found.**). The highest point of the basin is 3823 m and the lowest point is 1328 m with the average of 2377 m above sea level. The area of the basin is 216 km$^2$. The average
annual precipitation is 640 mm and the average annual discharge is 78.23 Mm$^3$ at Sulaghan station. The hydrometric stations include Kiga (Gage 1), Keshar (Gage 2) and Sulaghan (Gage 3) (Figure 1). As shown in Figure 1 three rain gage stations in the basin and four synoptic stations around the basin record the precipitation.

## 3 Methodology and data

In this study SOI, MEI, AO, NAO, and MJO teleconnection indices were evaluated to select an index that has the highest
correlation with the precipitation in the study area. In this paper, Fisher's exact test of independence was used to test the significance of correlation between the teleconnection indices and precipitation. In the Fisher's exact test the null hypothesis is that the two variables are independent. In other words, the relative proportions of each teleconnection index are independent of the precipitation:

$$H_0 : \rho = 0$$
$$H_1 : \rho \neq 0$$
(1)

Considering the Fisher's exact test, if p-value is less than 0.05 the null hypothesis is rejected; i.e. the p-value must be less than 0.05. According to the results (Table 1), there would be a statistically significant association between the SOI (MEI and NAO, as well) and Precipitation. However, SOI has the highest correlation to the Kan Basin precipitation. Therefore, ENSO is the most important large-scale atmospheric signal that affects Iran's climate. It has been shown that rainfall intensity increases in

the conditions of El-Niño and decreases in the conditions of La-Niña. Accordingly, the present study will assess the increase in flood damage due to El-Niño occurrence. For this purpose, the following steps have been taken.

**Step I:** Estimating the lag time ($T_l$) between the El-Niño event and the precipitation in the study area. As the effect of ENSO takes time to be experienced in far geographic locations, the lag time between the ENSO occurrence and the related influences in Kan River Basin was firstly calculated. This lag time can be revealed by comparison the variations of SOI and local precipitation time series. The monthly rainfall at the nearby synoptic stations of Mehrabad (1951-2017), Shemiran (1988-2017), Tehran-Geophysics (1992-2017) and Chitgar (1997-2017) (See Figure 1) and monthly SOI values are used. A statistical method, the average mutual information (AMI), is used to determine the time delay. This method is based on the Shannon entropy theory and is a measure of the "amount of information" obtained about one random variable, through the other random variable. Guiasu (1977) defined the mutual information of two random variables as a measure of the mutual dependence between two variables. Not limited to real-valued random variables and linear dependence like the correlation coefficient, mutual information is more general and determines how different the joint distribution of the pair (X,Y) is to the product of the marginal distributions of X and Y (Guiasu 1977). Suppose A is monthly precipitation in the representative station of the basin and B is the Southern Oscillation Index (SOI). The AMI is defined between two measurements $a_i$ and $b_i$ belonging to the sets A and B, respectively as follows (Cover and Thomas 1991):

$$I_{AB} = \sum_{i=1}^{K}\sum_{j=1}^{K} P_{AB}\left(a_i, b_j\right) \log\left(\frac{P_{AB}\left(a_i, b_j\right)}{P_A\left(a_i\right) P_B\left(b_j\right)}\right) \tag{2}$$

where $P_{AB}(a_i, b_j)$ is the conjugate probability density for measurements A and B with values of $a$ and $b$, respectively; $P_A(a_i)$ and $P_B(b_j)$ are the probability density function for measurements A and B. If $a_i$ (the measurements of A) is independent of $b_i$ (the measurements of B) then the value of $I_{AB}$ will be zero. In this paper, probabilities $P_A$ and $P_B$ were calculated using empirical frequency analysis in which the relative frequency histograms for both time series, SOI and precipitation were determined. The values of AMI for different arbitrary lag-times (1 to 12 months) between SOI and precipitation were calculated. The higher AMI value, the more dependency between two time series. Therefore, that lag-time corresponding to the highest AMI value was selected as the lag-time between the time series. This method determines the lag time between two time series by using the nature of the data itself and without any predetermined format based on probabilistic concepts (Cover and Thomas 1991).

In Eq. (2), $K$ is the optimal number of statistical categories for fitting the statistical distribution on the measurements A and B. When dealing with large sets of numbers, Sturge's rule (Sturges 1926) can be used to choose the number of categories. Sturge's rule is widely used in the statistical packages like excel for making histograms. According to Sturge's rule the data range should be split into $K$ equally spaced classes where:

$$K = 1 + 3.332 \log_{10}\left(n\right) \tag{3}$$

where $n$ is the number of data in the corresponding interval (here n = 480); therefore, in the present study, $K = 10$. Noted that there is uncertainty in the optimal number of categories that may influence the lag time between the precipitation in the basin and SOI.

**Step II:** Estimating the amount of rainfall variation under the influence of El-Niño: Secondly, the influence of El-Niño on the precipitation amount in Kan River Basin is quantified. The influence is estimated using a statistical method by calculating the expected value of the changes of precipitation amount in the El-Niño episodes compared to those in the neutral periods. Doing so, SOI a standardized index based on the observed sea level pressure differences between Tahiti and Darwin is applied. In this study SOI values were obtained from National Oceanic and Atmospheric Administration (NOAA) website (https://www.cpc.ncep.noaa.gov/data/indices/soi). Prolonged periods of negative SOI values accompany the abnormally warm ocean waters across the eastern tropical Pacific. La-Niña and El-Niño are characterized respectively by SOI>+0.8 and SOI<-0.8 (Australia Bureau of Meteorology 2012). Then, the "precipitation change" (PC) in the El-Niño condition and the average precipitation change ($\Delta P$) is calculated as follow:

$$\Delta P = \frac{\sum_{i=1}^{n} PC_i}{n} = \frac{\sum_{i=1}^{n} (P_{Eli} - P_N)/P_N}{n} \tag{4}$$

where $P_{El}$ is the annual rainfall in the El-Niño episode; $P_N$ average annual rainfall in the normal episodes; and $n$ is number of El-Niño events in the time period. PC values then will be used to construct synthesized rainfall storms for simulation of the El-Niño influence. It is a major limitation of this research that the annual change factor is applied in the extreme rainfalls of short time scales. Certainly, it was better to consider the monthly or seasonal change factor or calculation of change factor on the basis of recorded storms the applying it in a continues hydrologic model to have a more accurate prediction of El-Niño effect on the flood damages then calculation of the annual damages over years, but because of data limitation the analyses performed for the annual data. It is better to consider the uncertainties in a specific way. The employed probabilistic method for considering the rainfall increase percentiles can cover some of these uncertainties.

**Step III.** Estimation of design rainfalls with different return periods: Thirdly, several design storms are generated to be applied in a rainfall-runoff model. The rainfall storms are synthesized based upon the average precipitation change during El-Niño events. The designed storms are used for assessing the flood damages in a certain return period. To do so, the intensity-duration-frequency (IDF) rainfall curve of Kan basin is obtained from Water Research Institute (WRI 2011b). The time of concentration of the basin is $T_c$=135.19 min (Using Kirpich equation). Therefore, the intensity of the design rainfall can be deduced for different return periods. In this research, three return periods of 5, 10 and 50 years are considered for the rainfall storms. These values are selected in accordance with the paper's objective to show the importance of small floods in flood management plans compared to the high return period floods. As the El-Niño affects the rainfall intensity, then three distinct scenarios of El-Niño influence (normal, weak El-Niño and strong El-Niño conditions) were defined. To determine the rainfall intensity in every scenario, PC values are employed and using an appropriate analytic probability distribution, the rainfall

increase in different confident levels are determined. Here two probability levels, 60% and 90%, are considered for every rainfall return period. Accordingly, 9 different model runs were evaluated in the following scenarios:

- Scenario I (normal condition): In the first scenario no El-Niño event is considered. It is assumed that the basin receives a rainfall with the given intensities (*T*=5, 10, and 50) for a duration equal to *Tc*.

- Scenario II (weak El-Niño condition): In the second scenario it is assumed that the rainfall intensity increases at the 60% probability level.

- Scenario III (strong El-Niño condition): In the third scenario it is assumed that the rainfall intensity increases at the 90% probability level.

The reason behind the choice of 60% and 90% probability levels is to estimate the average amount of damages and the maximum amount of damages that are expected per year due to moderate to strong El-Niño events. Therefore, a probability level representative of the maximum possible damage and a probability level representative of average damage caused by El Niño were selected. Noted that, this is a limitation in our methodology in which the increase percentile for the rainfall for every return period has been considered the same.

**Step IV:** Hydrological modeling: Fourthly, the HEC-HMS hydrologic model is used to simulate the rainfall-runoff process. The hydrologic model is run for every scenario and every return period; then the peak discharges are used in the next step to estimate the flooding depths. In the hydrologic model, the SCS method is used to calculate the effective rainfall ($P_e$):

$$P_e = \frac{(P - 0.2S)^2}{P + 0.8S} \qquad\qquad (5)$$

where $P$ is rainfall; and $S$ is storage potential. The constant value 0.2 in Eq. (5) is selected based on SCS recommendation. SCS has proposed the initial losses can be estimated as $I_a$=0.2$S$ (Ponce and Hawkins 1996). We did not have any estimate for this losses therefore, the SCS recommendation was taken for Kan River Basin. Moreover, Clark instantaneous unit hydrograph method is applied to transform the effective rainfall into runoff ($Q$). Two-parameter Muskingum method is used for flood routing. The Muskingum method calculates the discharge within the river given the inflow hydrograph at the upstream end. For calibration of the HEC-HMS model, hourly historical storms which had been recorded in 3 rain gage stations in the basin and the related runoffs at the hydrometric stations (Figure 1) are used. Noted that for calculation of rainfall specified to every sub-basin, the gage weight method is used where the weights were determined from Thiessen method. The curve numbers (CN) and time of concentrations ($T_c$) are calibrated within the 10 sub-basins. For calibration and verification of the hydrologic model four storm events were extracted from 15 years available data (2000-2014): 1) the storm of 15–18 April 2003 in which a flood of maximum 38.22 m3/s was recorded at Gage3; 2) the storm of 16–19 April 2002 where the peak discharge rate of 32.3 m3/s was recorded at 10 Gage3, 3) the storm of 15–17 April 2009 in which a flood of maximum 34 m$^3$/s was recorded at Gage3 and 4) the storm of 11–13 March 2011 where the peak discharge rate of 55.1 m3/s was recorded at Gage3. Then the

calibrated model can be used for modelling the rainfall of given return periods to calculate the flood hydrographs at the outlet of the sub-basins.

**Step V:** Hydraulic modeling: Fifthly, the HEC-RAS model is used for hydraulic modeling and determination of flood depth at the target points (Residential areas shown in Figure 1). Based on the obtained flood depth, the flooding areas are determined for designed storms in the El-Niño and neutral periods. HEC-RAS is a one-dimensional model based on the numerical solution of the Saint-Venant equations. The model calibration is done by adjusting the Manning roughness coefficients at different river sections. The calibrated model then is used under steady state condition to model the obtained peak discharges from HEC-HMS in order to calculate flood depths at the target points.

**Step VI.** Estimating expected damage cost: Finally, flood damage is assessed for all 9 runs of the model. These damages can be compared to each other in order to determine the role of El-Nino on the flood damages. Damage caused by flooding can be divided into two groups: tangible and intangible damages. Intangible damages are those caused by illnesses and mental problems due to loss of life or properties. Tangible damage can be categorized into two direct and indirect damages. Direct damage is that caused from flooding of the buildings and properties such as home equipment, crops, livestock and poultry. Indirect damages are those caused due to disruption of trade and business, threating life and needs to emergency services, and so on. Noted that, this paper focuses on the direct tangible damages only.

Damage caused by flood is a function of its characteristics, including flow depth and inundation amount, duration of inundation, and flow velocity. One of the commonly used methods for estimating flood damages is damage-elevation curve method which gives the relationship between damage percentile and flood depth (Corry et al. 1980, KGS_Group 2000, Messner et al. 2007; Olesen et al. 2017; Wobus et al. 2017; Jamali et l. 2018). Damage-Elevation curves that are prepared for different land uses of Kan River Basin are presented in Figure 2. This method has been developed by the Federal Emergency Management Administration (Berkman and Brown 2015). The main land uses of the Kan River floodplain are residential buildings, restaurants, and fruit gardens (WRI 2011c). The damage cost then is calculated having entire monetary value of the inundated land uses. A comprehensive analysis of physical damages due to flooding requires many information including accurate updated land use map, area and age of buildings, type of the structure, number of floors, exact areas of different agricultural crop in the flood-prone area, crop number per unit area, value of crops, value of buildings (residential and non-residential) and their contents, number of residential, administrative, and commercial buildings in flood prone areas, the area and elevation of buildings, their locations, and spatial distribution of flood depth values in the inundated areas for different return periods. In this paper a simplistic approach is used for this regard. For the building damage analysis, separating residential and commercial ones, the total area of inundated buildings, average inundation depth, and the average economic value of the building and their contents for every buildings type are used. For agricultural damage analysis, considering the dominant crops of cherry and apple, the area of inundation, average inundation depth, crop density, and average price of one single crop the flood damage costs are evaluated.

## 4 Results and discussion

Monthly analyses of the precipitation in the synoptic stations and SOI using AMI method showed that there is no lag time between rainfall and SOI time series (the lag time is less than one month).

In Figure 3, the annual rainfall of stations is plotted against the SOI index. It is obvious that with decreasing SOI index, annual rainfall increases in the study area and vice versa. In the period of 1951 to 2017, a total of 9 El-Niño (SOI<-0.8) and 7 La-Niña (SOI>+0.8) events have been occurred. Out of them, 6 years have experienced increase in the precipitation and 3 years with decrease in the precipitation. The largest event for El-Niño dates back to 1983 and 1987 with respectively 334 mm and 252 mm recorded rainfall in Mehrabad station. Furthermore, based on the trendlines, in average one unit decrease in the SOI, will enhance 22.5 mm annual rainfall in Mehrabad station. For further analyses, Mehrabad station was chosen because it has more data than the other stations.

Evaluation of the monthly SOI over the period from 1951 to 2017 shows that 197 months with El-Niño, 163 months with La-Niña, and 432 normal months have been occurred. The average monthly rainfall at Mehrabad station in the months of El-Niño is 22.2 mm and in the months of La-Niña 16.96mm, while in the normal months, the average rainfall is 18.41 mm. Therefore, using Eq. (4) El-Niño increases the rainfall amount in the study area by 20.6% and La-Niña decreases it by 7.86%.

But in the risk analysis, it is required to evaluate the annual damage costs. To evaluate the annual damage costs, under the effect of El-Niño, at first the years with El-Niño condition (SOI<-0.8) were recognized. There are 9 years with El-Niño and 50 normal years among the total of 66 years (1951-2017). Then, using Eq. (4), PC and ΔP can be calculated. According to the results, for 9 years with El-Niño condition, PC ranges from -60.34% to 42.8% while the latter is related to the year 1983 in which 334 mm rainfall was recorded. On the basis of Kolmogorov-Smirnov goodness of fit test with 99% certainty, Gumbel distribution well fits on these percentiles (Figure 4). the median in Figure 4 is close to zero (4%) and the mode value (for which the probability density function is maximum) is 10.1%. In fact, the reason for that the median is close to zero is due to the selection criterion of 9 events as the El-Niño events out of total events; (SOI less than -0.8 according to Australia Bureau of Meteorology). If this criterion is set as SOI<-1.0 (according to the Western Regional Climate Center, USA), the median increases to 12.2; because two El-Niño events with negative %-increased precipitation are eliminated. As the same way, if the criterion be changed as SOI< -1.02, the median will increase significantly to about 20. Therefore, the criterion for distinguishing the El-Niño condition an effective assumption in this paper and the results of Figure 4 should not be evaluated as the insignificant effect of El-Niño on the Kan River Basin precipitation.

For probabilistic analysis of flood damage costs, two probability level of 60% and 90% were considered. The first (60% confidence interval) is representative of average amount of damages (a weak El-Niño condition) and the other (90% confidence interval) is representative of maximum amount of damages (a strong El-Niño condition). These levels respectively show a moderate effect of El-Niño and a high effect of El-Niño on the annual damage costs of flood.

According to Figure 4, it can be said that by 90% and 60% certainty the increased percentiles of rainfall during El-Niño years is less than 31% and 8.2% respectively compared to the normal years. Based on these results, three scenarios of increase rate

of rainfall are investigated. In the first scenario (normal condition) rainfall is assumed to occur in the normal condition without any increase in the rainfall amount. In the second scenario (weak El-Niño condition), rainfall is assumed to increase by 8.2% (60% confidence interval), and in the third scenario (strong El-Niño effect), an increase of 31% rainfall is supposed to be taken place due to El-Niño (90% confidence interval).

The Kan River basin has 135.19 min time of concentration. Therefore, considering the duration of the design rainfall as $D$=150 min, the rainfall intensity ($i_d$) can be estimated from IDF curves. For return periods of 5, 10, and 50-yr, $i_d$ values are 7.8, 9.5, and 13 mm/hr, respectively. Thus, in the first scenario, three storms with the abovementioned intensities during D=150 min are modeled. In the second and third scenarios it is assumed that with the same duration, total rainfalls increased by 8.2% and 31%, respectively.

**4.1 Hydrologic and hydraulic modeling**

HEC-HMS is calibrated using an actual rainfall and runoff event recorded in April 2003. The calibration parameters include curve number and concentration time ($T_c$) of the sub-basins. In the process of automatic calibration, the parameters are determined such a way that the model could simulate the hydrologic behaviour of the basin accurately. The main objective is to predict the exact peak discharge and time to peak of the hydrograph in the hydrometric stations by minimizing the mean

squared error (MSE) between the predicts and the observations. Comparison the observed and simulated hydrographs in Gage 3 is illustrated in Figure 5a. Moreover, Figure 5b shows the results for the upstream gages (Gage2 and Gage3). In Table 2, the calibration result of the hydrologic model is presented. Then the hydrologic model is verified with the storm event in April 2002, April 2009 and March 2011 resulted in floods with respectively peak discharges of 31.5 m³/s, 34.4 m³/s and 54.1 m³/s. Comparison between the simulated and observed flood hydrographs are shown in Figure 5c. It is noted that, for the flood of

11-13 March 2011, the peak of 54.1 m3/s has been estimated by Regional Water Company of Tehran. It is noted that, for these flood events, the discharges are not available in the upstream stations. As a result, the model can be used for modelling the design storms in the three scenarios to calculate the flood hydrographs at the sub-basins.

Then HEC-RAS software is used to simulate the flood travelling and to determine the flood depths of different return periods in the target points. For starting the hydraulic modelling, HEC-RAS requires cross sections of the river in different points. In

this study the cross sections were extracted from Digital Elevation Model. Flood hydrographs of the sub-basins with different return periods were simulated by the calibrated rainfall-runoff model (HEC-HMS) and the peak values were used as the boundary conditions for the HEC-RAS model. For the model's calibration, the peak discharges produced in the hydrologic model's calibration step (flood 15–18 April 2003) are input into the hydraulic model as the boundary conditions at the upstream reaches and the flood depth and discharge at Sulaghan station is compared with the observed one. The calibration parameters

are Manning roughness coefficients that are calibrated manually. For the model verification, flood in 16–19 April 2002 and the upstream peak discharges generated in the hydrologic model are used.

Therefore, the hydraulic model can be used for calculating the flow depth at the target points. The hydrologic and hydraulic models are applied for modelling the design storms of different 5, 10, and 50 return periods under three defined scenarios. It means that totally 9 different runs of the sequent models are required. Using the obtained inundated depths, the flood mapping can be done in ArcMAP. Figure 6 illustrates the inundation areas at two target points (Sulaghan and Sangan villages) for the 50-yr flood in the first scenario.

## 4.2 Damage analysis

In this section, with the help of GIS tool and the land use maps (generated from maps of 1/25000 scale) which were obtained from the local municipality, a simple analysis of damages to the buildings, their contents, and agricultural areas is carried out. In this step just five Sub-basins of Imamzadeh Davood, Rendan, Sangan, Sulaghan and Keshar are considered; because of lack of land use maps, low population, and low development in the other sub-basins. As noted previously, here a simplistic approach with several limitations is used. The main reason for this is the lack of precise land use information and accurate spatial value of each property in the basin. At the basin level, for example, there is a lot of gardens along the river, some with less than one-year-old trees and some with more than 20 years old trees that have different economic values and various vulnerability to the flood. In this article, all such gardens are seen in the same way. Moreover, it is supposed that all the agricultural land is used for apple and cherry because other fruit gardens include of very low area in the basin. Furthermore, in the basin there are buildings of one to three floors with different areas, some of them are new and some are old; therefore, they are not of equal value and same vulnerability to flood. While in this article all buildings are considered similar and the damage cost to them was estimated by total area of buildings in the inundation area.

For this regard, applying the inundation map on the land use maps, the average depth of inundation and area of inundation for every land use are calculated. Then from the damage-elevation curves percentile of damage to the land uses can be estimated. Finally the damage cost to each of land uses is calculated by the average economic value of one unit of that land use. It should be noted that for agricultural physical damages analysis in every sub-basin, two dominant products of cherry and apple were identified and based on the percentage of each of them, average crop number per unit area and value of each crop, the damage analysis was performed. Percentages of crops, number of them per unit area and their economic value as well as value of different assets in the flooding area is obtained by a field survey and interviews with the local authorities and local inhabitants, and engineering judgment. Table 3 provides the details of the physical damage costs to the present land uses for the 50-year flood. Similarly, two scenarios of rainfall increase are simulated and the damages are estimated in the same procedure. Table 4' shows the amount of flood damages cost for different return periods in the three considered scenarios. This table revealed that, firstly, the expected flood damages cost during El-Niño event increases much more than that of rainfall increase, and secondly, in the smaller return periods, the increases of flood damages is much more than that in the bigger return periods. In the return period of 5-year, with 60% probability El-Niño may increase the damages up to 1072%, although the expected increase for the rainfall is 8.2%. Also, with 90% probability the increased damages cost is less than 2959% compared to the

normal condition. The main reason for this high amount is that the average depth of 5-yr flood is very small (<0.04m) and with an increase of 8.2% or 31% in the rainfall amount, the flood depth increases considerably (respectively less than 0.41m and 0.63m). During the 10-yr flood, due to El-Niño, increases up to 133% (by 60% probability) and 289% (by 90% probability) are expected to be seen in the flood damages cost. Similarly, for the return period of 50-year, the increase in the probable flood losses would drop to 41% and 74% at the probability level of 60% and 90%, respectively

It should be noted that, the high %-increase for 5-yr return period may relate to the fact that the initial losses and infiltration amounts are considerable compared to the 5-year rainfall. In such condition, effective rainfall is so reduced that it produces not too much runoff. For higher return periods while the rainfall amount rises but the event duration does not change, infiltration and initial losses increase slightly and, in total, the runoff increases significantly. It is clear that if the increase in precipitation is significant enough, then the runoff increase will also be significant. Therefore, in the cases of large return periods in which the rainfall amounts are large, the infiltration and initial losses are not significant, and so the increase in rainfall intensity will be mainly seen in the increase in the effective precipitation and direct runoff.

## 5 Conclusions

In the present paper, the effect of El-Niño on the probable flood damages was investigated. The methodology was based on the calculation of increasing rainfall amount at 60% and 90% confidence intervals during El-Niño event compared to the normal conditions. It should be noted that to have a comprehensive study on the risk of flood due to El Nino, it is required to consider the whole range of probabilities, but due to the objectives of the paper the lower tail of distribution was not considered in the risk analysis. The increase percentile of the rainfall was then applied for generating design storms of different return periods of 5, 10 and 50 years. Here, three scenarios were defined: 1) normal condition scenario; 2) rainfall increase scenario because of El-Niño at 60% confidence level (equal to 8.2% increase in the rainfall intensity); and 3) rainfall increase scenario at 90% confidence level (31% increase in the rainfall intensity due to El-Niño). Noted that the median %-increased rainfall due to El-Niño is about 4% and taking all range of PC probabilities consisting the lower tail of distribution where rainfall during El-Niño years is actually decreased and then numerically integrate over the distribution will be close to the median of the distribution. Using HEC-HMS and HEC-RAS models flood zoning was performed in the three defined scenarios and three return periods, as well. Therefore, a total of 9 models were developed and flood zoning results turned into physical damage. To estimate the flood damage cost the damage-elevation curve method was used. The results showed that the occurrence of El-Niño in less return periods, which is more frequent, increases the damages very much and for the higher return periods the increase percentile drops considerably. More specifically in a flood with a return period of 5 years, with a 90% chance an increase of 2959% may be occurred. The average increase in the expected damages cost is 1072% for 5-yr return period, while it is 133% and 41% for 10-yr and 50-yr return periods, respectively. It is remarkable that rainfall intensity increases by 8.2% and 31% in the second and third scenarios. Consequently, it implies that the flood management in this basin should pay more attention to the small floods during the El-Niño years. In general, flood management in small flood case requires much less

financial budget and may result in much more effective approaches. These results is a warning to decision makers to take in to account the probable effect of El-Niño on the programs of flood risk management. These results provide decision makers with essential information on flood risk and highlight the importance to take in to account the probable effect of El-Niño in flood risk management

## 5 Acknowledgments

This research is partially supported by the National Natural Science Foundation of China (41730645, 41790424, 41425002), the Strategic Priority Research Program of the Chinese Academy of Sciences (XDA20060402), and the International Partnership Program of Chinese Academy of Sciences (131A11KYSB20170113). Qiuhong Tang is supported by the Newton Advanced Fellowship.

## 10 Conflict of interest

The authors declare no conflict of interest.

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

**Table Captions:**

Table 1 Correlation coefficient (r) and p-value between precipitation in Mehrabad station and different teleconnection indices

Table 2: Calibration results of the hydrologic model

Table 3: Physical damages to the sub basins properties for 50-yr flood in different scenarios

5    Table 4: Flood damages cost and expected increases during El-Niño event

**Figure Captions:**

**Figure 1:** Location of Kan River Basin in north of Tehran plain, Iran

**Figure 2:** Damage-elevation curves for different land uses of a) Building and its contents, b) Restaurant's content, and c) agriculture in Kan River Basin.

**Figure 3:** Annual rainfall against SOI index in the station of a) Mehrabad, b) Shemiran, c) Chitgar and 4) Tehran Geophysics

**Figure 4:** Gumbel Cumulative distribution function fitted on the annual precipitation changes

**Figure 5:** The observed and simulated flood hydrographs in a) calibration step at Sulaghan Station (15–18 April 2003); b) calibration step at upstream gages (15–18 April 2003); and c) verification step at Sulaghan Station(16–19 April 2002, 15–17 April 2009, and 11–13 March 2011)

**Figure 6:** The 50yr floodplain in the sub-basins of a) Sulaghan and b) Sangan (Original layers are available in WRI 2011a)

30

**Table 1**

| Index | SOI | MEI | NAO | AO | MJO |
|---|---|---|---|---|---|
| r | 0.32 | 0.29 | 0.15 | 0.002 | 0.1 |
| $p$-value | 0.016 | 0.02 | 0.029 | 0.84 | 0.9 |

30

**Table 2**

| No. | Sub-basin | Area (km²) | Calibration parameter | |
|---|---|---|---|---|
| | | | $CN$ | $Tc$ (hr) |
| 1 | Imamzadeh Davood | 23.77 | 71.87 | 1.05 |
| 2 | Rendan | 33.61 | 72.35 | 0.874 |
| 3 | Sangan | 47.43 | 71.37 | 1.227 |
| 4 | Taloon | 26.65 | 71.39 | 0.932 |
| 5 | Kiga | 4.40 | 70.2 | 0.332 |
| 6 | Doab | 7.19 | 71.71 | 0.511 |
| 7 | Keshar | 34.85 | 71.83 | 1.181 |
| 8 | Herias | 11.44 | 70.7 | 0.759 |
| 9 | Sulaghan | 13.66 | 71.1 | 0.556 |
| 10 | Jangalak | 12.89 | 71.9 | 0.623 |

| First scenario | | | | | |
|---|---|---|---|---|---|
| Sub-basin | Average inundation depth (m) | Damage to residential building ($10^3$US$) | Damage to content ($10^3$US$) | Damage to restaurant ($10^3$US$) | Damage to agriculture (US$) |
| **Imamzadeh Davood** | 0.61 | 201 | 37 | 42 | 519 |
| Rendan | 0.6 | 68 | 12 | 20 | 723 |
| Sangan | 0.5 | 393 | 68,084 | 19 | 1,259 |
| Keshar | 0.66 | 343 | 62 | 67 | 550 |
| Sulaghan | 0.25 | 81 | 11 | 55 | 843 |
| Sum ($10^3$US$) | | **1,086** | **190** | **203** | **3,893** |
| Total damage cost ($10^3$US$) | | | | | **5,372** |
| Second scenario | | | | | |
| Imamzadeh Davood | 0.76 | 246 | 49 | 51 | 715 |
| Rendan | 0.7 | 816 | 15 | 24 | 935 |
| Sangan | 0.66 | 455 | 82 | 22 | 2,098 |
| Keshar | 0.82 | 483 | 92 | 95 | 715 |
| Sulaghan | 0.45 | 176 | 40 | 119 | 1,080 |
| Sum ($10^3$US$) | | **1,441** | **277** | **311** | **5,543** |
| Total damage cost ($10^3$US$) | | | | | **7,572** |
| Third scenario | | | | | |
| Imamzadeh Davood | 0.89 | 324 | 61 | 67 | 832 |
| Rendan | 0.79 | 88 | 18 | 26 | 1,028 |
| Sangan | 0.76 | 517 | 102 | 25 | 2,398 |
| Keshar | 0.89 | 514 | 98 | 101 | 739 |
| Sulaghan | 0.65 | 298, | 54 | 201 | 1,844 |
| Sum ($10^3$US$) | | **1,741** | **333** | **421** | **6,840** |
| Total damage cost ($10^3$US$) | | | | | **9,334** |

Table 4

| T (yr) | Damage cost (US$) | | | Damage increase (%) | |
|---|---|---|---|---|---|
| | 1th scenario (no El-Niño effect) | 2nd scenario (8.2% rainfall increase) | 3rd scenario (31% rainfall increase) | 2nd scenario | 3rd scenario |
| 5 | 120,529 | 1,413,100 | 3,032,458 | 1,072 | 2,416 |
| 10 | 1,393,753 | 3,256,633 | 5,149,147 | 133 | 269 |
| 50 | 5,372,472 | 7,571,796 | 9,334,019 | 41 | 74 |

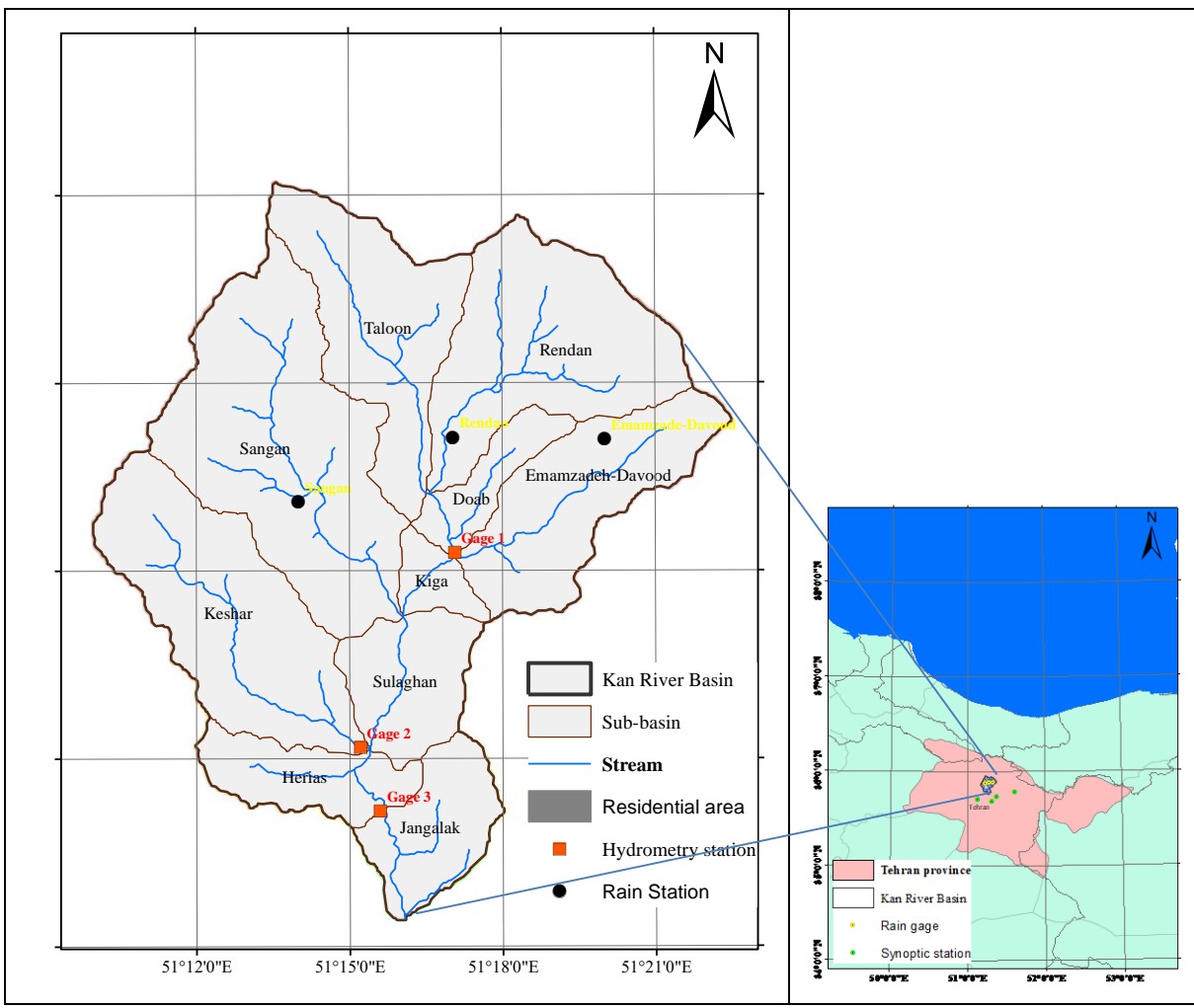

**Figure 1**

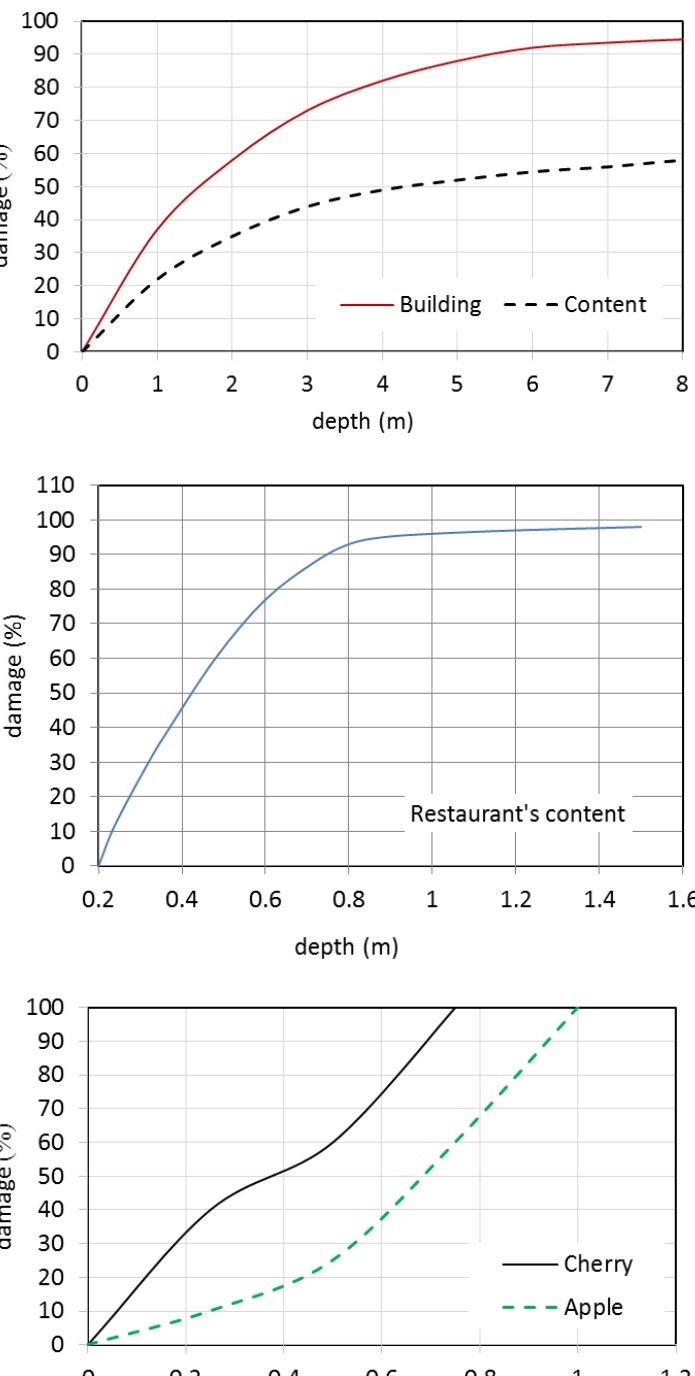

**Figure 2**

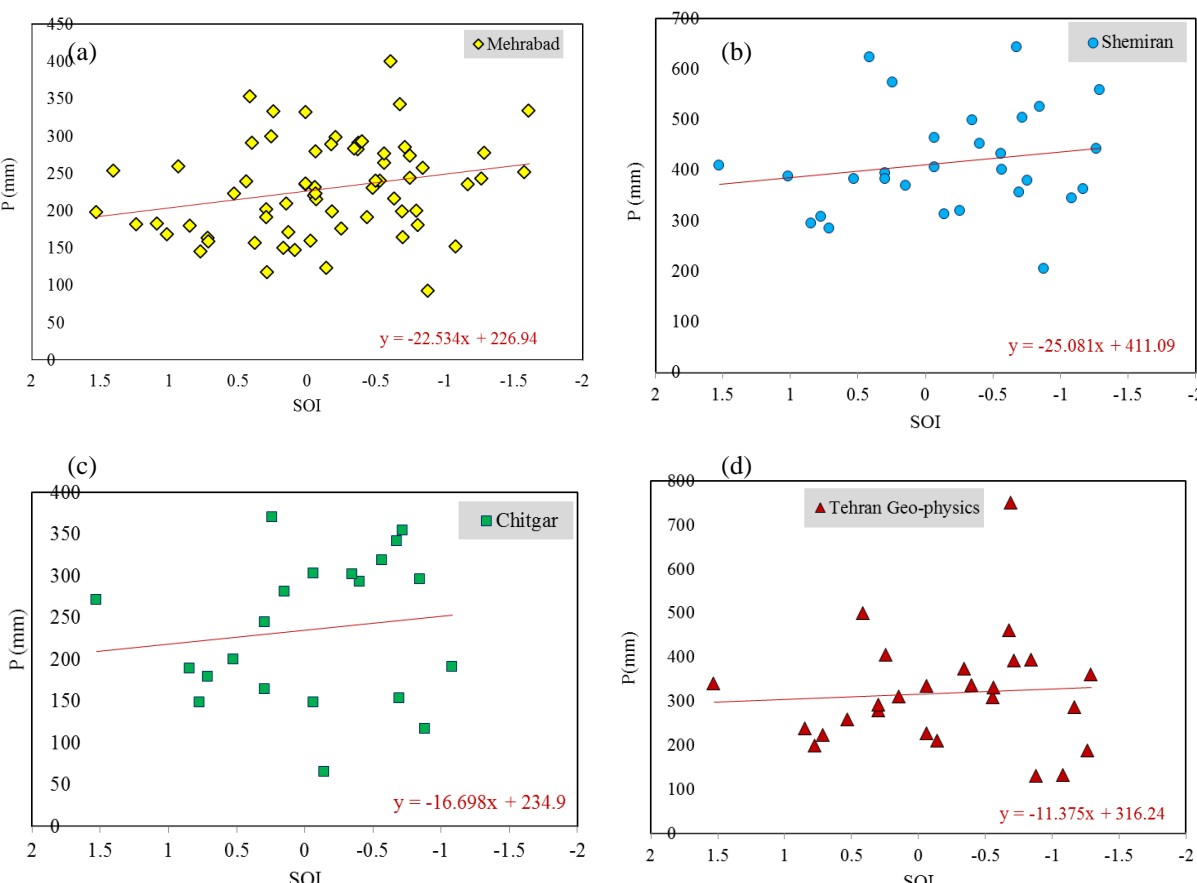

5                                                    **Figure 3**

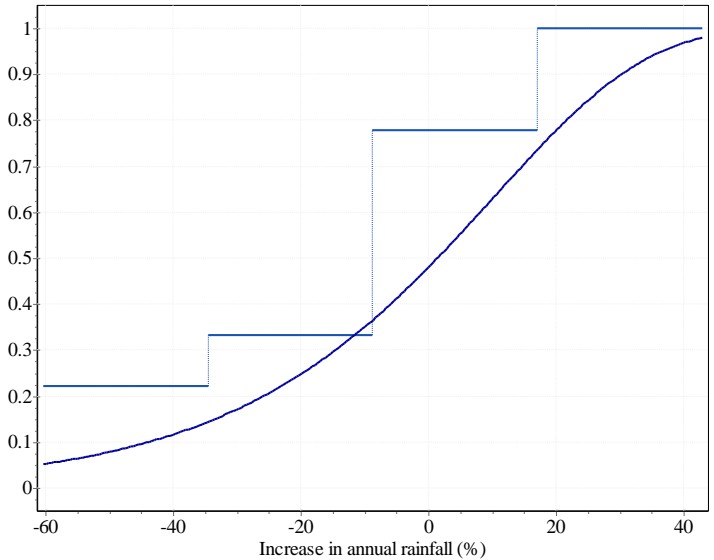

**Figure 4**

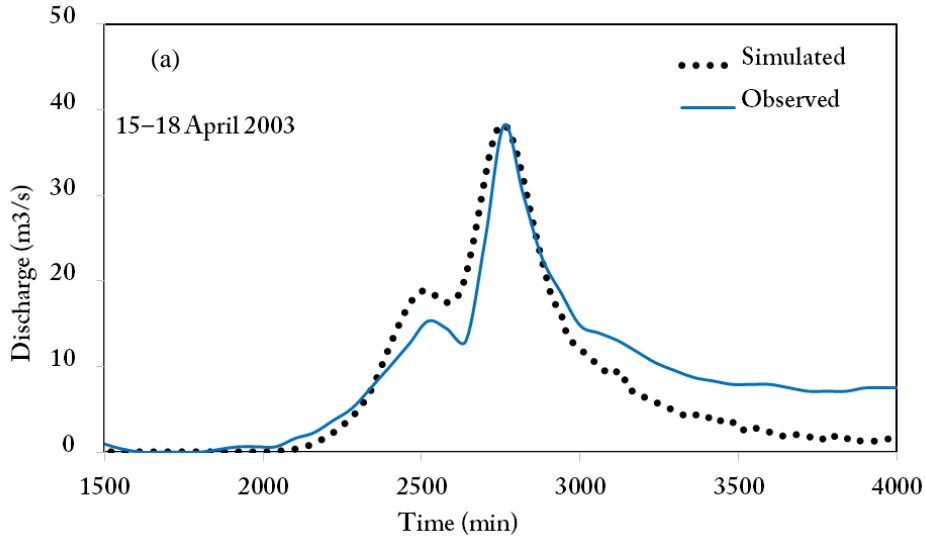

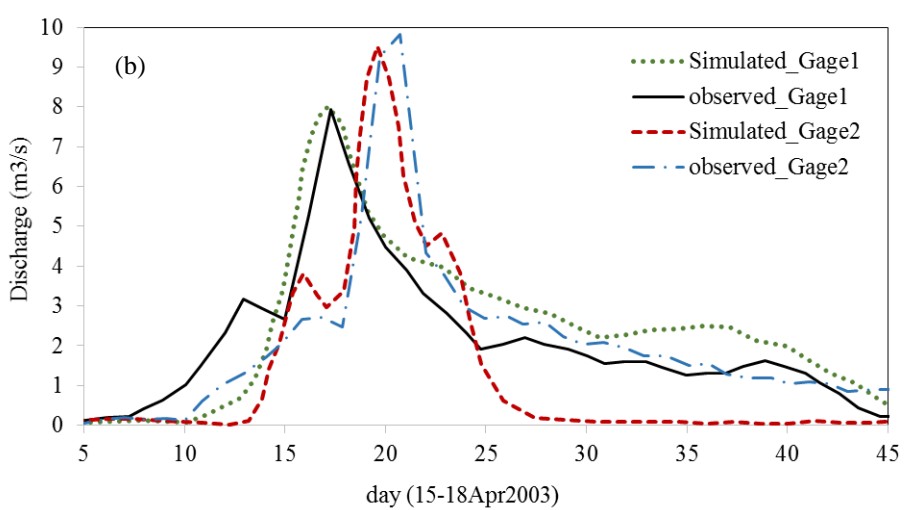

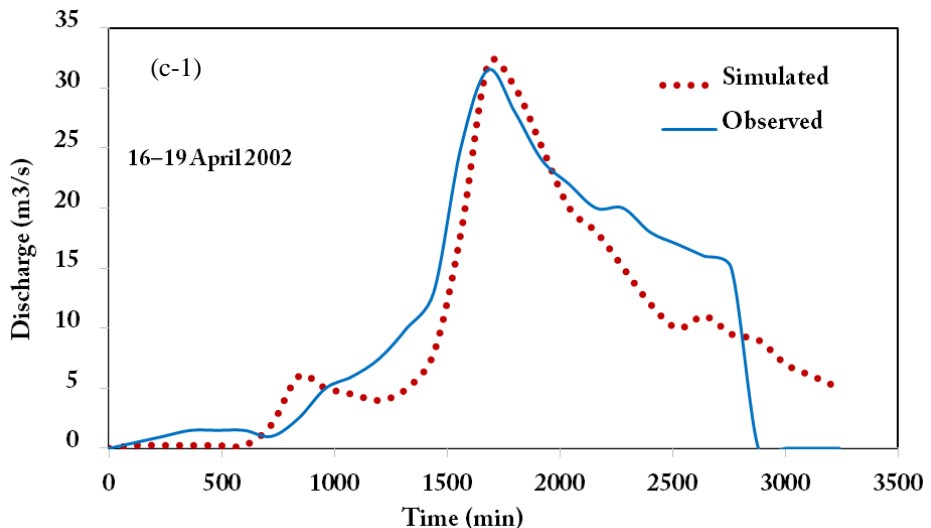

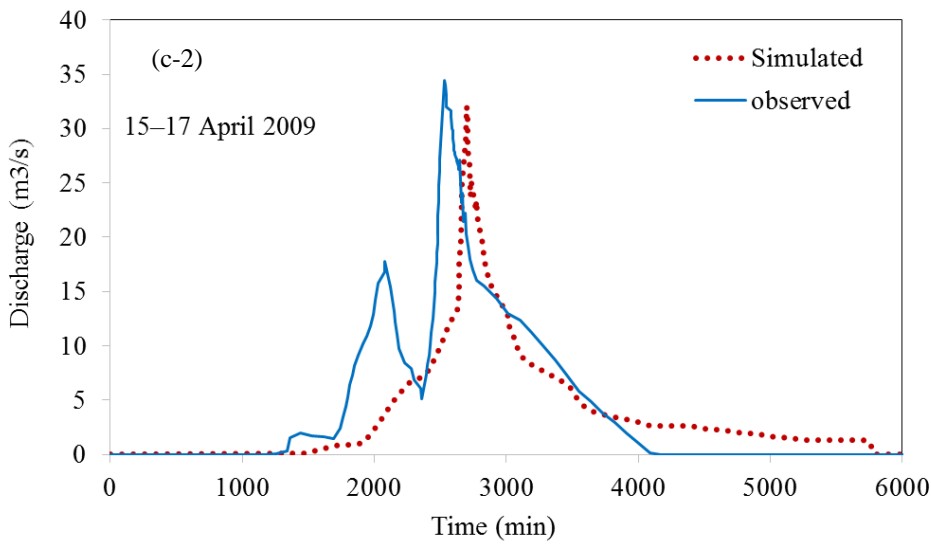

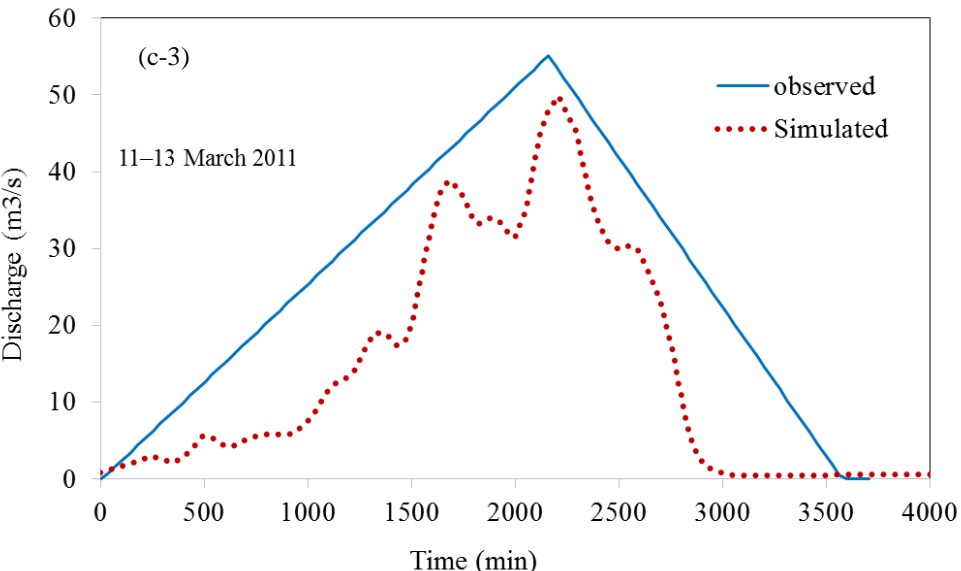

**Figure 5**

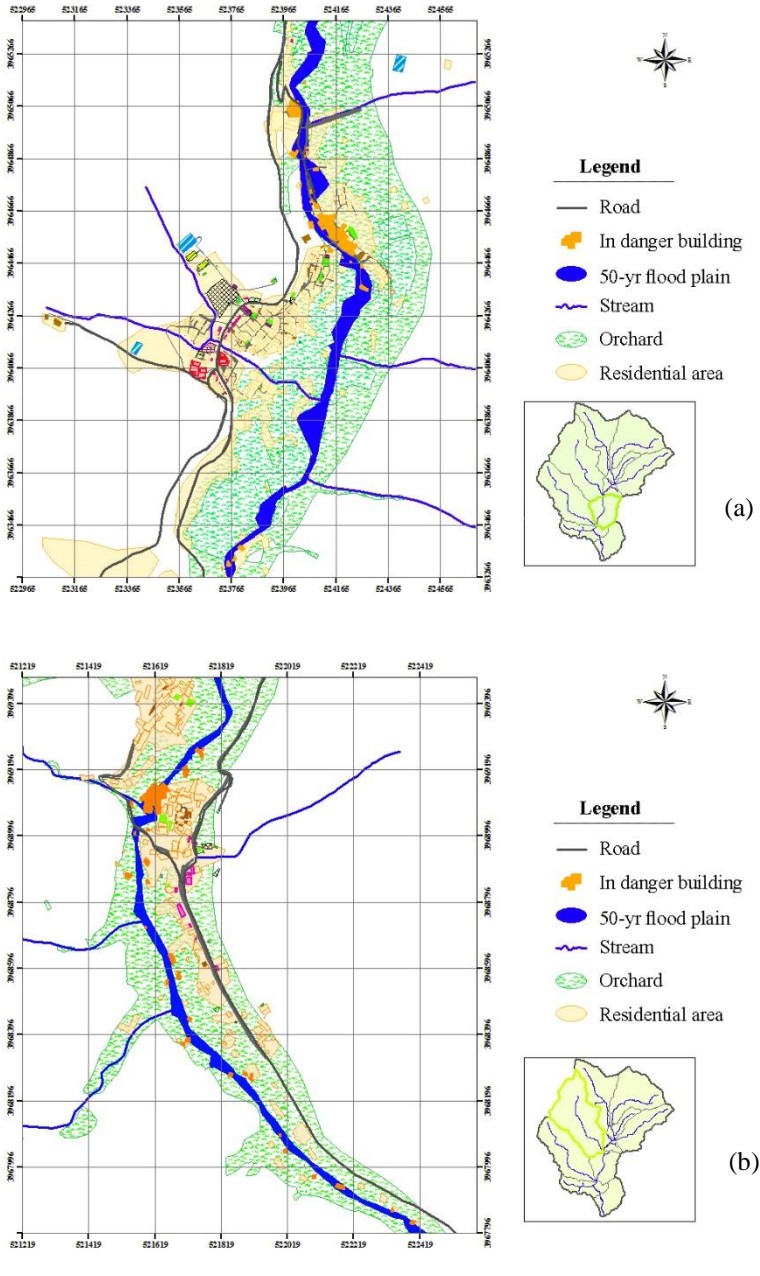

**Figure 6**