# Peer review of "Evaluation of the probable annual flood damage influenced by El-Niño in the Kan River Basin, Iran"

_Natural Hazards and Earth System Sciences, 2019_

## Referee Comment (RC1) · Anonymous Referee #1 · 23 Aug 2019

The authors present an investigation of the impact of El Nino on flood damage. The analysis is carried out by (1) establishing the average impact of El-Nino and La-Nina on the precipitation measured in a station within the study area; (2) based on this average impact, different rainfall scenaria are run through a hydrologic model to calculate inundation depths for different return periods and (3) a simple loss model is applied to quantify the damage due to the different inundation scenaria.

I think the topic is important and of broad interest because, as the authors claim, only a few studies have focused on the impact of large-scale climatic phenomena on flood damage and losses. However, I do not believe this analysis, in its current state, is

suitable for publication. I found the following major issues with the study:

1) The impact of El-Nino and La-Nina on precipitation, as measured by equation (3), is an average impact. It is unlikely that rainfall extremes, such as those considered in the study for return periods (RP) of 10, 25, and 50 years, are impacted in the same way. Therefore, it is inaccurate to multiply the 10, 25, and 50 year RP rainfall values by the same, average factor in order to account for the impact of El-Nino and La-Nina. More generally, I do not think it is appropriate to assume that the impact of El-Nino and La-Nina on annual precipitation totals is the same as that on single precipitation events, which is, I believe, what the authors are doing.

2) The impact of ENSO on precipitation varies by season (e.g., Alizadeh-Choobari and Najafi, 2018), so it is a bit simplistic to reduce the impact of ENSO to an annual average, especially when this average impact is then applied to single events.

3) The hydrologic model is calibrated using a single event (April 2003). I think the calibration period is too short (only 4 days) to be meaningful, the model should be calibrated for longer periods, to account, for example, for antecedent conditions and seasonality of the hydrologic cycle. I do not see why the authors did not do that as data seem to be available.

4) The manuscript is not well written, and its lack of clarity makes it very difficult to follow and understand. I've reported below many different instances in which sentences are wrong or not clear. I suggest the authors to carefully check the manuscript before re-submission.

SPECIFIC COMMENTS:

P2L25: Since this is the topic of the paper, it would be good to cite some of these works in which the expected damage under El Nino or La Nina is investigated.

P3L23: SOI is a single variable ENSO index, multivariable ENSO indexes like MEI provide a more complete description of ENSO (e.g. Wolter and Timlin, 2011). Can the

authors justify why they are not using a multivariable ENSO index instead?

P5L5to10: It is not clear how the calibration has been done. Do the authors calibrate using the storm in 2003 and validated using the storm in 2002? If so it seems like a very small calibration and validation set, especially when 15 years of data seem to be available.

P6L4: It does look like there is a small trend, however I would not say it is "obvious" as there is a lot of data scatter. It would be good to provide a level of significance for the trend.

P6L6: "with respectively 334mm and 252mm recorded rainfall", are these annual rainfall totals? It sounds like that but in Figure 2 for Mehrabad station I don't see a point for 334mm with SOI>0.8.

P4L24and26: This is the first time the values 60% and 90% are introduced in the manuscript. The authors should say why they are focusing specifically on these numbers as it is not clear.

P6L4toL19: time scales (annual or event scale) are a bit unclear in this part.

P6L7: "its sufficient data", what makes it sufficient? Perhaps it can be said that Mehrabad station was chosen because it has more data than the other stations.

P6L9: Here the time period is 1952-2017 while before it was 1950-2017.

P6L11: "22.2m and 16.96mm" is this average monthly rainfall?

P6L13toL19: This part is very unclear and should be significantly expanded: What are the "annual increased percentiles of rainfall"? How are they computed using Eq.3? Why are the increased percentiles of rainfall in Fig.3 only 4 values? Can't the authors use more points to compute the percentiles?

P6L23: I struggle to understand why 60 and 90 percentiles are considered confidence intervals. Can the authors please explain?

P7L1toL14: As before, I don't understand why only two events are used, one for calibration and another for validation when the authors have many more data at their disposal.

P721: It would be helpful to know the size of these land use areas. Averaging flood depth and economic values in large areas can give very uncertain results because both flood depth and economic value usually have very large variability in space.

P7L24: Are the "flood zoning maps" the flood maps the authors obtained with Hec Ras? It is not clear whether the authors produce flood maps with Hec Ras and then apply the maps on their land use areas or if the flood zoning maps are predefined.

P7L24: How is the "average depth of land uses' inundation" defined? It sounds like the authors calculate an average flood depth for each land use which would make sense to me, but then in table 2 they seem to have only one average flood depth common to the different land uses which is even more simplistic. Why can't the authors calculate an average flood depth for each land use? that would be more accurate I believe.

P8L3toL4: I think such a big increase in loss warrants a deeper explanation. Why does an 8.2% increase in rainfall produces a 10-fold increase in flood depth?

MINOR COMMENTS

P1L25: have increased or have been increasing.

P1L27: reached to doesn't make sense.

P2L8: "...by predicting the necessary measures", it is not clear what you mean by "measures".

P3L2: rain spouted 6 times of the annual precipitation is not clear.

P3L4: "events" not "evens" and I think they mean "that resulted", not just "resulted".

P3L14: It should be: "According to previous studies" or "according to Hooshyaripor et

al. (2018)".

P4L3: Perhaps you should specify that in Eq. 2 log is the logarithm to the base 10.

P4L9,10: Sentence should be rephrased.

P4L23: If I understood correctly it should be "for a duration equal to Tc" not "in Tc min duration".

P4L25: "Affected by".

P4L24and26: These sentences are very difficult to understand. Do you mean that the rainfall intensity with Tc duration is increased by 60% and 90%? If so, why are you talking about probability? If not, could you please rephrase the sentences in a more understandable way?

P5L23: Do you mean duration of the inundation or the time at which the inundation occurred? The former is more likely I believe.

REFERENCES:

Alizadeh-Choobari O, Najafi MS (2018) Climate variability in Iran in response to the diversity of the El Niño-Southern Oscillation. Int J Climatol 38 (11):4239–4250.

Wolter, K., and M. S. Timlin ( 2011), El Niño/Southern Oscillation behaviour since 1871 as diagnosed in an extended multivariate ENSO index (MEI.ext), Intl. J. Climatol., 31, 1074– 1087.

---

## Referee Comment (RC2) · Anonymous Referee #2 · 26 Sep 2019

This study aims to assess the impact of El-Nino on expected flood damages. The idea is relevant and novel, while the execution and documentation currently are not sufficient for publication.

**1   Major comments**

Reading the paper I noted similar major issues as RC1:

1. In large parts of their methodology the authors jump to simplistic approaches.

[Figure]

This may in some cases be necessary because no better knowledge is available. However, the authors do not provide any arguments / literature references in this direction and also don't use data analysis to reason for their approach. I included details below.

2. The documentation is incomplete in some parts and it is not always clear what data were used.

3. In addition to the above, I have some difficulty understanding the motivation for this work. Clearly, atmospheric circulation patterns must be expected to impact extreme rain intensities. However, considering sufficiently long rainfall time series, these oscillations should not affect the probability distribution of extreme rainfall and thus also not our estimates of expected flood damages? Why then do we need to know exactly how much damages vary over time?

**2  Methodological issues**

1. Linking rainfall series and SOI - The references I found describe the AMI as a univariate method and I could not find it in the reference provided by the authors. It is not clear how the joint probabilities are computed from the histograms and the equation for the "optimal number of categories" appears out of the blue. Most importantly, the authors did not provide any evidence that the method gives reasonable results (time series plots of SOI and rainfall indicating the identified lag, scatterplots of lag vs. AMI, cross correlation plots, or similar)

2. Determining increased rainfall during El Nino

    (a) There is an obvious problem in using change factors derived for annual rainfall to extreme daily/hourly precipitation. No reasoning is provided for why this is done.

(b) I believe Fig. 3 illustrates the %-changes of rainfall for all years that were identifed as "El Nino" in step 1.

    i. It is not clear to me why you would pick the 60 and 90% quantiles for the further analysis. The median value, as far as I can see, is 0. My conclusion would be that there is no evidence for an impact of El Nino on annual rainfall? Also the trends identified in Fig.2 look very questionable. Have you tested the significance of parameters?

    ii. Are the annual rainfall and %-change values correlated? This would certainly impact the trend estimates in Fig. 2 and pose a challenge for distribution fitting in Fig. 3?

3. Hydrological and hydraulic modelling

(a) The hydrological model was calibrated for a single event only, which is not good practice. Why is only one of the 3 stations used for calibration?

(b) It is not clear for which areas the HEC-RAS simulation is performed (not highlighted in Fig.1), so I cannot evaluate whether the link between hydrological and hydraulic modelling setup makes sense.

4. Damage calculation - It is not clear how the damage calculation was performed. The depth-damage functions are not provided in the paper. In the results section, the authors mention the computation of an "average inundation depth" per landuse class. This seems like a questionable approach, but it is simply not clear what was done here.

**3  Minor comments**

P1L27: How do you define a flood event? Is it correctly understood that Tehran experienced flooding 12 times in 1951 and 54 times in 1991?

P2L1-23: This part of the introduction cites a lot of studies that measured impacts of atmospheric circulation patterns. However, most of these refer to completely different parts of the world, so I had difficulty seeing the relevance.

P3L10: typo 78.23M cbm/s ?

P4L20-25: What is a natural uniform rainfall?

---

## Referee Comment (RC3) · Anonymous Referee #3 · 1 Oct 2019

The manuscript described an interesting idea on looking at the flood damage caused by ENSO. A case study in Kan River Basin, Iran was presented. The manuscript is well organized and easy to follow. However, I think the current results are not convincing enough, because huge uncertainties from the 6-step were ignored. Extrapolation without acknowledging uncertainties can mislead the result. Thus, I recommend return the manuscript to authors for major revision.

Major comments:

In methodology section, authors presented a 6-step method to investigate the flood damage from ENSO. In each of those steps, there are uncertainties. In particular, the

steps I, II, III and VI are based on probabilistic models, in which uncertainties cannot be avoid. The main issue is that when putting those 6 steps in series the uncertainty can be exploded. This makes the result meaningless when having huge uncertainty.

When estimating the relationship between rainfall variation and SOI, large uncertainty should exist on the slope as shown in Figure 2. The results derived from Eq 3 should also have a large uncertainty. When bringing this uncertainty into next steps, the final results may be very different from what has been found right now.

Minor comments:

There are some constant values in equation 2 and 4. Authors need to justify these numbers.

---

## Author Comment (AC1) · 13 Dec 2019

**Response Letter to Reviewers Comments on nhess-2019-166**

Many thanks for the quick response from the editor and the reviewers. The manuscript has been improved substantially based on the constructive comments of the reviewers.

**(The highlighted parts are added to the revised paper)**

**Response to Comments of Reviewer #1**

**Comment #1:** The authors present an investigation of the impact of El Nino on flood damage. The analysis is carried out by (1) establishing the average impact of El-Nino and La-Nina on the precipitation measured in a station within the study area; (2) based on this average impact, different rainfall scenario are run through a hydrologic model to calculate inundation depths for different return periods and (3) a simple loss model is applied to quantify the damage due to the different inundation scenario. I think the topic is important and of broad interest because, as the authors claim, only a few studies have focused on the impact of large-scale climatic phenomena on flood damage and losses. However, I do not believe this analysis, in its current state, is suitable for publication. I found the following major issues with the study:

The impact of El-Nino and La-Nina on precipitation, as measured by equation (3), is an average impact. It is unlikely that rainfall extremes, such as those considered in the study for return periods (RP) of 10, 25, and 50 years, are impacted in the same way. Therefore, it is inaccurate to multiply the 10, 25, and 50 year RP rainfall values by the same, average factor in order to account for the impact of El-Nino and La-Nina. More generally, I do not think it is appropriate to assume that the impact of El-Nino and La-Nina on annual precipitation totals is the same as that on single precipitation events, which is, I believe, what the authors are doing.

**Response**: In the present paper a risk base analysis has been performed and the increase amounts of damage costs are not deterministic. Indeed, the results of %-damages enhancements that are reported in the paper, do not mean that in every El Nino concurrent with a storm event the flood damages will certainly be increased as those calculated in the paper but it considers a chance for every %-damage increase. On this basis, results of Table 3 mean that for example if a 10-yr return period rainfall is happening while a strong El Nino condition is experiencing, it is expected by 90% probability that the damages cost be less than 267% compared to the normal condition.

The second issue is that the rainfall and damage enhancements that are presented in the paper are the expected increase values of yearly rainfall and damages which have been calculated from a long time series of data. Therefore, the %-changes of damages represent the expected annual values for every return period or the values that are probable by the given certainty levels. **Figure 1** illustrates the Log-Logistic distribution that is fitted on the %-changes of

rainfall in January and April. Noted that, according to the results El Nino in average increases the rainfall in April and decreases it in April. This figure shows that in 60% certainty the increased rainfall in Tehran is less than 36% in January and in April it follows not only an increased amount but a decreased value less than -28%. However, in the annual time scale which is reported in the paper for the same probability the rainfall increase is less than 8.2%.

[Figure]

**Figure 1: cumulative distribution function fitted on the increased percentiles of rainfall in**

**a) January and b) April**

It is obvious that the rainfall increase and the relevant flood damage for a specific month can be higher than that we have calculated in the yearly time scale and vice versa for another month (the yearly change percentile is the monthly averaged for 12 months). Indeed, during a year with El Nino condition, some months experience El Nino, some months experience La

Nina and the others experience neutral episode, but the El Nino months are dominant. Therefore, although in El Nino months it is expected that the flood damages to be increased, in the La Nina and normal months they are expected to be reduced and fixed, respectively. Having an average over the monthly %-changes of damage costs in a year, we could have the annual %-changes of damages which expect to be near to those percentiles that are presented in the paper. The same procedure, but more and more time consuming, can be followed for the time scales less than month if there are appropriate enough data of precipitation and SOI. Unfortunately, there is no long time continues daily data of precipitation in the basin for conducting such analysis. Kan Basin although it is a small basin with limited hydrologic available data, because it is very close to Tehran city with about 10 million population has a great importance regarding to flood management. Kan River Basin provides a busy recreational and pilgrimage place for the citizens from the close cities in the weekends and holydays to spend their time near the river. During past decades, due to flood events many people have died in the basin. The last catastrophic flood disaster returns to June 2015 in which 8 people died, although many other flood events have been recorded, as well. Therefore, in spite of lack of data, the importance of the basin forces the researchers to study the risk of flood and to present the flood mitigation measures.

**Comment #2:** The impact of ENSO on precipitation varies by season (e.g., Alizadeh-Choobari and Najafi, 2018), so it is a bit simplistic to reduce the impact of ENSO to an annual average, especially when this average impact is then applied to single events.

**Response**: With great thanks; we read and used the paper: Alizadeh-Choobari and Najafi (2018). Your point is completely correct and as explained in the previous comment we confirm that the seasonal flood damages will be more than that annual ones that we have reported in the paper. However, having an average over the seasons in a year, the annual flood damages can be calculated.

Average impact considering single events: It should be noted that our results show that in total the maximum increase in flood damages would be expected if for example a 10-yr return period rainfall happens while the El Nino condition is experiencing. It is not total annual flood damages or an average of yearly damages. The %-damages could be viewed as the probable values regarding the average annual rainfall increase in the basin during El Nino condition.

**Comment #3:** The hydrologic model is calibrated using a single event (April 2003). I think the calibration period is too short (only 4 days) to be meaningful, the model should be calibrated for longer periods, to account, for example, for antecedent conditions and seasonality of the hydrologic cycle. I do not see why the authors did not do that as data seem to be available.

**Response**: This paper aimed at evaluating the flood influence on the societies in term of physical damages while ENSO can affect the precipitation amount, therefore we needed to have a hydrologic model calibrated for flood events to give us acceptable estimates for peak

discharges; i.e. we did not need for a continues hydrologic model in a year. On this basis, we looked for the main historical flood events in the available meteorological and hydrological time series. According to the data, most of the floods in the study area have taken place in March to May. In the available time series, we had problem sometimes because of missing precipitation data, sometimes missing runoff data and sometimes the concurrency of rainfall storm with the recorded runoff.

During 2002 to 2016, the basin has experienced several flood events and we have used all the events with observations for the model calibration and verification. In addition to the flood of 15–18 April 2003 and 16–19 April 2002 which have been mentioned in the manuscript, two other events are also have been considered for calibration and verification of the model. Floods of 2009 and 2011 with peak discharges of 34.4 $m^3$/s and 54.1 $m^3$/s. These results are added to the paper.

Also two other storm events are also have been considered for verification of the hydrologic model. Floods of 2009 and 2011 with peak discharges of 34.4 $m^3$/s and 54.1 $m^3$/s. Comparison between the simulated and observed flood hydrographs are shown in Figure 2 and Figure 3. It is noted that, for the flood of 11-13 March 2011, the peak of 54.1 $m^3$/s has been estimated by Regional Water Company of Tehran.

[Figure]

**Figure 2:** Observed and simulated flood hydrographs at Sulaghan Station in 15–17 April 2009

[Figure]

**Figure 3:** Observed and simulated flood hydrographs at Sulaghan Station in 11–13 March 2011

**Comment #4:** The manuscript is not well written, and its lack of clarity makes it very difficult to follow and understand. I've reported below many different instances in which sentences are wrong or not clear. I suggest the authors to carefully check the manuscript before re-submission.

**Response:** Thanks for the comment. The paper is totally checked, revised and all the suggestions are considered carefully in the revised paper.

**SPECIFIC COMMENTS**

**Comment #5:** P2L25: Since this is the topic of the paper, it would be good to cite some of these works in which the expected damage under El Nino or La Nina is investigated.

**Response:** It is done and the introduction part was improved by more relevant papers.

Although, the effect of ENSO on the precipitation has been frequently studied in Iran (Nazemosadat and Ghasemi 2004; Saghafian et al. 2017; Alizadeh-Choobari et al. 2018; Hooshyaripor et al. 2018), there are few studies about ENSO influence on the socioeconomic impacts of floods even around the world (Ward et al. 2014). The main reason for the limited research on the economic impacts of climate and hydrologic variability is said to be the lack of economic data on flood damages (Changnon 2003). Analyzing the National Flood Insurance Program daily claims and losses and Multivariate ENSO Index (MEI), Corringham and Cayan (2019) quantified insured flood losses across the western United States from 1978

to 2017. They showed that in coastal Southern California and across the Southwest of the United States, El Niño has had a strong effect in producing more frequent and higher magnitudes of insured losses, while in the Pacific Northwest, the opposite pattern with weaker and less spatially coherent has been reported. Changnon (2003) revealed that the strong El Niño events of 1982/83 and 1997/98 have caused significant flood damages over \$2.8 billion in Southern California. Null (2014) demonstrated that from 1949 until 1997 out of the six seasons that flood damages costs exceeded \$1 billion in California three cases had been El Niño years; one very strong (1982), one moderate (1994) and one weak (1968). Ward et al. (2014) showed that ENSO exerts strong and widespread influences on both flood hazard and risk. They assessed ENSO's influence in terms of affected population, gross domestic product and economic damages on the flood risk at the global scale and showed that climate variability, especially from ENSO, should be incorporated into disaster-risk analyses and policies. They revealed that, if the frequency and/or magnitude of ENSO events were to change in the future due to climate change, change in flood-risk variations across almost half of the world's terrestrial regions is happened. Ward et al. (2016) provided a global modelling exercise to examine the relationships between flood duration and frequency and ENSO. They indicated that the duration of flooding compared to flood frequency is more sensitive to ENSO.

Changnon, S. 2003. Measures of economic impacts of weather extremes. Bull. Amer. Meteor. Soc., 84, 1231–1235, https://doi.org/10.1175/BAMS-84-9-1231.

Corringham, T.W. and Cayan, D.R. 2019. The Effect of El Niño on Flood Damages in the Western United States, Weather, Climate, and Society, 11(3), 489-504. https://doi.org/10.1175/WCAS-D-18-0071.1.

Null, J. 2014. El Niño and La Niña: Their Relationship to California Flood Damage, Golden Gate Weather Services, August 2014.

Ward, P.J., Jongman, B., Kummu, M., Dettinger, M.D., Sperna Weiland, F.C. and Winsemius, H.C. 2014. Strong influence of El Niño Southern Oscillation on flood risk around the world, Proceeding of National Academy of Sciences of America (PNAS), 111(44), 15659-15664.

Ward P.J. Kummu M., Lall U. 2016. Flood frequencies and durations and their response to El Niño Southern Oscillation: Global analysis, J Hydrology, Volume 539, August 2016, Pages 358-378.

**Comment #6:** P3L23: SOI is a single variable ENSO index, multivariable ENSO indexes like MEI provide a more complete description of ENSO (e.g. Wolter and Timlin, 2011). Can the authors justify why they are not using a multivariable ENSO index instead?

**Response:** In this study several indices were evaluated to find those that have the highest correlation with the precipitation. Doing so, SOI, MEI, AO, NAO, and MJO were analyzed, finally the results showed that SOI has the highest correlation coefficient with the precipitation in Kan River Basin even a little more than MEI. In Figure 4 you can find the annual precipitation against the aforementioned indices. Based on the highest correlation between the

annual precipitation and teleconnection indices, we picked the SOI for further analysis. A summary of these results has been added in the revised paper.

In this study SOI, MEI, AO, NAO, and MJO teleconnection indices were evaluated to select an Index that has the highest correlation with the precipitation in the study area. Results are shown in Table 1. According to the results, there would be a statistically significant association between the SOI (MEI and NAO, as well) and Precipitation. However, SOI has the highest correlation to the Kan Basin precipitation. Therefore. SOI was selected for further analysis.

TABLE 1 CORRELATION COEFFICIENT (R) AND P-VALUE BETWEEN PRECIPITATION IN MEHRABAD STATION AND DIFFERENT TELECONNECTION INDICES

| Index | SOI | MEI | NAO | AO | MJO |
|---|---|---|---|---|---|
| r | 0.32 | 0.29 | 0.15 | 0.002 | 0.1 |
| $p$-value | 0.016 | 0.02 | 0.029 | 0.84 | 0.9 |

[Figure]

[Figure]

[Figure]

[Figure]

**Figure 4**. Scatter plot of the annual precipitation in Mehrabad Station against the teleconnection indices

**Comment #7:** P5L5to10: It is not clear how the calibration has been done. Do the authors calibrate using the storm in 2003 and validated using the storm in 2002? If so it seems like a very small calibration and validation set, especially when 15 years of data seem to be available.

**Response**: The validation is limited by the number of flooding events with observations. We have used all the observations available. This comment was also replied in Comment # 3 in detail.

**Comment #8:** P6L4: It does look like there is a small trend, however I would not say it is "obvious" as there is a lot of data scatter. It would be good to provide a level of significance for the trend.

**Response:** In this paper Fisher's exact test of independence was used to test the significance of correlation between the teleconnection indices and precipitation in Kan River Basin. In the Fisher's exact test the null hypothesis is that the two variables are independent. In other words, the relative proportions of each teleconnection index are independent of the precipitation:

$H_0 : \rho = 0$

$H_1 : \rho \neq 0$

Considering the Fisher's exact test, if $p$-value is less than 0.05 the null hypothesis is rejected; i.e. the $p$-value must be less than 0.05. Table 1 summarizes the $p$-value and correlation coefficient values for different indices examined here. According to Table 1, there would be a statistically

significant association between the SOI (MEI and NAO, as well) and Precipitation. However, SOI has the highest correlation to the Kan Basin precipitation.

**Comment #9:** P6L6: "with respectively 334 mm and 252 mm recorded rainfall", are these annual rainfall totals? It sounds like that but in Figure 2 for Mehrabad station I don't see a point for 334mm with SOI>0.8.

**Response**: These values are the total annual precipitation. El-Niño condition relates to SOI<-0.8 and La-Niña condition relates to SOI>+0.8. The precipitation of 334 mm and 252 mm have been taken place during El-Niño not La-Niña, therefore these precipitation should be looked for in SOI<-0.8 which are shown Figure 5 (Fig 2 in the paper).

[Figure]

**Figure 5.** Annual precipitation in Mehrabad station against the SOI

**Comment #10:** P4L24and26: This is the first time the values 60% and 90% are introduced in the manuscript. The authors should say why they are focusing specifically on these numbers as it is not clear.

**Response**: Figure 6 (Fig 3 in the paper) shows the cumulative distribution of precipitation increases (%) in the El Niño years compared to normal years. Toward a risk-based analysis to the flood damage resulting from El Niño, the probability of any precipitation and its increase percentile occurring in the long-term time series of the study area is important. For this reason, Figure 6 has been developed and used to determine the requirements.

In this article it was required to estimate the average amount of damages and the maximum amount of damages that are expected per year. Therefore, a probability level representative of the maximum possible damage and a probability level representative of average damage caused by El Niño were selected. According to Figure 6, if we divide the precipitation increase into

two classes of zero to 20% and 20 to 40%, the 60% probability level can represent the mean precipitation increase and the 90% probability level can be considered as the maximum precipitation increase.

[Figure]

**Figure 6:** Gumbel Cumulative distribution function fitted on the annual increased percentiles of rainfall

**Comment #11:** P6L4toL19: time scales (annual or event scale) are a bit unclear in this part.

**Response:** In this part, annual precipitation has been analyzed. We have made the text clear for time scale in the revised manuscript.

**Comment #12:** P6L7: "its sufficient data", what makes it sufficient? Perhaps it can be said that Mehrabad station was chosen because it has more data than the other stations.

**Response:** You are right. It is corrected as follow:

Mehrabad station was chosen for further analyses because it has more data than the other stations.

**Comment #13:** P6L9: Here the time period is 1952-2017 while before it was 1950-2017.

**Response**: According to data of the Mehrabad station 1951-2017 is correct (66 year data). So it is corrected in the revised version.

**Comment #14:** P6L11: "22.2m and 16.96mm" is this average monthly rainfall?

**Response**: Of course, it is monthly averaged.

**Comment #15:** P6L13toL19: This part is very unclear and should be significantly expanded: What are the "annual increased percentiles of rainfall"? How are they computed using Eq.3? Why are the increased percentiles of rainfall in Fig.3 only 4 values? Can't the authors use more points to compute the percentiles?

**Response**: This part was rewritten completely. We used the term "precipitation change" and "average precipitation change" to clarify the methodology instead of "increased percentiles of rainfall".

To evaluate the annual damage costs, under the effect of El-Niño, at first the years with El-Niño condition (SOI<-0.8) were recognized. There are 9 years with El-Niño and 60 normal years among the total of 66 years (1951-2017). Then, using Eq. 3, the "average precipitation change" (ΔP) was calculated according to the annual precipitations in the El-Niño conditions compared to those in the normal conditions. On the other hand, for every year with El-Niño, "precipitation change" (PC) can be calculated. For 9 years with El-Niño condition, the PC ranges from -60.34% to 42.8%. Fitting analytic probability distribution on these percentiles, Gumbel distribution can well be fitted on the basis of Kolmogorov-Smirnov goodness of fit test in 99% confidence interval. Figure 3 illustrates the Gumbel cumulative distribution function (CDF) fitted on the PC values. According to Fig. 3, it can be said that by 90% and 60% certainty the PC values during El-Niño compared to the normal years are less than 31% and 8.2%, respectively.

**Comment #16:** P6L23: I struggle to understand why 60 and 90 percentiles are considered confidence intervals. Can the authors please explain?

**Response**: It was explained in the 2 previous comment. Indeed, one of them (60% confidence interval) is representative of average amount of damages (a weak El-Niño condition) and the other (90% confidence interval) is representative of maximum amount of damages (a strong El-Niño condition). Both of them are possible to be happened in different probability levels. We wanted to show the moderate effect of El-Niño and high effect of El-Niño on the annual damage costs of flood. It is clear that the damages due to a moderate El-Niño, are more probable than the strong El-Niño; although the damages cost are lower.

For probabilistic analysis of flood damage costs, two probability level of 60% and 90% were considered. The first (60% confidence interval) is representative of average amount of damages (a weak El-Niño condition) and the other (90% confidence interval) is representative of maximum amount of damages (a strong El-Niño condition). These levels respectively show a moderate effect of El-Niño and a high effect of El-Niño on the annual damage costs of flood. According to Fig. 3, it can be said that by 90% and 60% certainty the increased percentiles of rainfall during El-Niño years is less than 31% and 8.2% respectively compared to the normal years.

**Comment #17:** P7L1toL14: As before, I don't understand why only two events are used, one for calibration and another for validation when the authors have many more data at their disposal.

**Response**: This comment was replied in comment #3 in detail.

**Comment #18:** P721: It would be helpful to know the size of these land use areas. Averaging flood depth and economic values in large areas can give very uncertain results because both flood depth and economic value usually have very large variability in space.

**Response**: The employed land use map is a vector shape file compose of polygons for different land uses that were generated from maps of 1/25000 scale. In this analysis we have used a simplistic method for damage costs evaluation. Indeed, the damage cost has been calculated having the average flood depth for every land use and the monetary value of one unit area of the inundated land use categories. Although the inundation depth is varied in the floodplain, the economic values are not significantly varied because the case study is a rural area with relatively the same life style in different places. Therefore, the economic value of one unit area of every land use category has been considered the same and the inundation depth has been averaged for each of categories.

**Comment #19:** P7L24: Are the "flood zoning maps" the flood maps the authors obtained with Hec Ras? It is not clear whether the authors produce flood maps with Hec Ras and then apply the maps on their land use areas or if the flood zoning maps are predefined.

**Response**: The flood depth is calculated in HEC-RAS and the inundation maps are generated in GIS. The flood map then is applied on the land use map for damage calculation. We ourselves have developed the models in our previous works. The flood zoning maps for 5, 10, and 50-yr return periods considering the increased rainfall intensities were developed specifically in the current paper.

**Comment #20:** P7L24: How is the "average depth of land uses' inundation" defined? It sounds like the authors calculate an average flood depth for each land use which would make sense

to me, but then in Table 2 they seem to have only one average flood depth common to the different land uses which is even more simplistic. Why can't the authors calculate an average flood depth for each land use that would be more accurate we believe.

Response: Certainly, we have calculated the damage cost base upon the average inundation depth for every land use category. The inundation depth in the table is the average of all inundation and as it is misleading we omit it from the table.

Comment #21: P8L3toL4: I think such a big increase in loss warrants a deeper explanation. Why does an 8.2% increase in rainfall produces a 10-fold increase in flood depth?

Response: Due to 8.2% and 32% increase in 5-yr rainfall intensity, the average increase in the average inundation depths are 35.4% and 64.8%, respectively not a 10-fold increase. However, the damage costs have been increased more than 10 times.
About the reason for this %-increases, it is noticeable that although %-increase in damages for the return period of 5 Year (a small return period) is high (1072% in scenario 2) for the bigger return periods the %-increase in flood damages is much smaller (for 10 and 50 return periods respectively 133% and 41% increase in scenario 2). The high %-increase for 5-yr return period may relate to the fact that the initial losses and infiltration amounts are considerable compared to the 5-year rainfall. In such condition, effective rainfall is so reduced that it produces not too much runoff. For higher return periods while the rainfall amount rises but the event duration does not change, infiltration and initial losses increase slightly and, in total, the runoff increases significantly. It is clear that if the increase in precipitation is significant enough, then the runoff increase will also be significant. Therefore, in the cases of large return periods in which the rainfall amounts are large, the infiltration and initial losses are not significant, and so the increase in rainfall intensity will be mainly seen in the increase in the effective precipitation and direct runoff.

**MINOR COMMENTS**

Comment #22: P1L25: have increased or have been increasing.

Response: changed to "Have been increasing".

Comment #23: P1L27: reached to doesn't make sense.

Response: ... from 12 cases in 1951 increased to 54 cases in 1991

Comment #24: P2L8: ". . .by predicting the necessary measures", it is not clear what you mean by "measures".

**Response**: ... Prediction of teleconnection indicators helps to reduce the flood damages by ==implementing the necessary practical measures==

**Comment #25:** P3L2: rain spouted 6 times of the annual precipitation is not clear.

**Response**: In June 1968 a heavy rain ==as 6 times as== the annual precipitation ==happened== in 2 days had caused 31 losses of life in Tehran central area and huge damage to the properties

**Comment #26:** P3L4: "events" not "evens" and I think they mean "that resulted", not just "resulted".

**Response**: In general, during a period of 60-year (from 1954 to 2015) at least 8 flood ==events that resulted== in loss of life (in total 2200 people) have been reported in Kan and 5 central Tehran areas.

**Comment #27:** P3L14: It should be: "According to previous studies" or "according to Hooshyaripor et al. (2018)".

**Response**: According to ==Hooshyaripor et al. (2018)== ENSO is the most important large-scale atmospheric signal that affects Iran's climate.

**Comment #28:** P4L3: Perhaps you should specify that in Eq. 2 log is the logarithm to the base 10.

**Response**: Of course yes. It is corrected

$$K = 1 + 3.332 \log_{10}(n)$$

**Comment #29:** P4L9,10: Sentence should be rephrased.

**Response**: It has been rephrased as:

==La-Niña and El-Niño are characterized respectively by SOI>+0.8 and SOI<-0.8 (Australia Bureau of Meteorology 2012).==

**Comment #30:** P4L23: If I understood correctly it should be "for a duration equal to Tc" not "in Tc min duration".

**Response**: Yes. We have now corrected it as: "==for a duration equal to Tc=="

**Comment #31:** P4L25: "Affected by".

**Response**: We have now corrected it as: "Affected by"

**Comment #32:** P4L24and26: These sentences are very difficult to understand. Do you mean that the rainfall intensity with Tc duration is increased by 60% and 90%? If so, why are you talking about probability? If not, could you please rephrase the sentences in a more understandable way?

**Response**: We have rephrased as:

- Scenario II (weak El-Niño condition): In the second scenario it is assumed that the rainfall intensity increases at the 60% probability level.
- Scenario III (strong El-Niño condition): In the third scenario it is assumed that the rainfall intensity increases at the 90% probability level.

**Comment #33:** P5L23: Do you mean duration of the inundation or the time at which the inundation occurred? The former is more likely I believe.

**Response**: We have now corrected it as: "duration of inundation"

**Comment #34:** REFERENCES:

Alizadeh-Choobari O., Najafi M.S. 2018. Climate variability in Iran in response to the diversity of the El Niño-Southern Oscillation. Int. J. Climatol. 38(11):4239–4250.

Wolter, K., and Timlin M.S. 2011. El Niño/Southern Oscillation behaviour since 1871 as diagnosed in an extended multivariate ENSO index (MEI.ext), Intl. J. Climatol., 31: 1074–1087.

**Response**: We have carefully read and use theses references in the paper.

---

## Author Comment (AC2) · 13 Dec 2019

**Response Letter to Reviewers Comments on nhess-2019-166**

Many thanks for the quick response from the editor and the reviewers. The manuscript has been improved substantially based on the constructive comments of the reviewers.

(The highlighted parts are added to the revised paper)

**Response to Comments of Reviewer #2**

**Major comments**

**Comment #1:** In large parts of their methodology the authors jump to simplistic approaches. This may in some cases be necessary because no better knowledge is available. However, the authors do not provide any arguments / literature references in this direction and also don't use data analysis to reason for their approach. I included details below.

**Response**: The basin is located north of Tehran and it is very important as described in the case study part. There are several rain gage and hydrometric stations in the basin and several synoptic station around the basin. Therefore, monthly meteorological and hydrologic data and information are sufficient; although there is lack of exact information about the spatial value of properties in the basin. Therefore, the simplistic approach we mentioned in the paper relates to the damage cost analysis and there is no shortage of meteorological and hydrological data in the study basin.

P6L1-13: "In fact, the simplistic approach that is used in this article is about damage estimation. The main reason for this is the lack of precise land use information and accurate spatial value of each property in the basin. At the basin level, for example, there is a lot of gardens along the river, some with less than one-year-old trees and some with more than 20 years old trees that have different economic values and various vulnerability to the flood. In this article, all such gardens are seen in the same way. Moreover, it is supposed that all the agricultural land is used for apple and cherry because other fruit gardens include of very low area in the basin. Furthermore, in the basin there are buildings of one to three floors with different areas, some of them are new and some are old; therefore, they are not of equal value and same vulnerability to flood. While in this article all buildings are considered similar and the damage cost to them was estimated by total area of buildings in the inundation area".

**Comment #2.** The documentation is incomplete in some parts and it is not always clear what data were used.

**Response**: In the revised paper, we explained the material and data everywhere it is required and where you mentioned in the comments.

a) Hydrologic model: The main input to the HEC-HMS model is rainfall. We used historical rainfalls but those recorded events that the corresponding runoff events in the hydrometric stations are available. CN and Tc values in all the sub-basins were chosen for calibration of the model. The objective function in the calibration step was to predict the exact peak discharge and time to peak of the hydrograph in the hydrometric stations by minimizing the mean squared error (MSE) between predicts and observations.

P5L5-10: "For calibration of the HEC-HMS model, hourly historical storms which had been recorded in 3 rain gage stations in the basin and the related runoffs at the hydrometric stations (Fig. 1) are used". Noted that for calculation of rainfall specified to every sub-basin, the gage weight method is used where the weights were determined from Thiessen method. The curve numbers (CN) and time of concentrations (Tc) are calibrated within the 10 sub-basins. For calibration and verification of the hydrologic model four largest storm events were extracted from 15 years available data (2000-2014): 1) the storm of 15–18 April 2003 in which a flood of maximum 38.22 m3/s was recorded at Gage3; 2) the storm of 16–19 April 2002 where the peak discharge rate of 32.3 m3/s was recorded at 10 Gage3, 3) ; and 4) ."

P7L4-8: "The main objective is to predict the exact peak discharge and time to peak of the hydrograph in the hydrometric stations by minimizing the mean squared error (MSE) between predicts and observations (Fig. 4a in the manuscript). In Table 1, the calibration result of the hydrologic model is presented. Then the hydrologic model is verified with the storm event in April 2002 (Fig. 4b in the manuscript) and finally used for modelling the design storms in the three scenarios to calculate the flood hydrographs at the sub-basins".

b) Hydraulic model: for starting the hydraulic modelling, HEC-RAS requires cross sections of the river in different points. In this study the cross sections were extracted from Digital Elevation Model. The boundary layers were defined as peak discharges at upstream reaches (output of the HEC-HMS) and the normal depth at downstream outlet.

P7L10-14: "For starting the hydraulic modelling, HEC-RAS requires cross sections of the river in different points. In this study the cross sections were extracted from Digital Elevation Model. For the model's calibration, the peak discharges produced in the hydrologic model's calibration step (flood 15–18 April 2003) are input into the hydraulic model as the boundary conditions at the upstream reaches and the flood depth and velocity at Sulaghan station are compared with the observed ones. The calibration parameters are Manning roughness coefficients that are calibrated manually. Noted that, for downstream boundary condition the flow depth at the outlet point was determined as the normal depth. For the model verification, flood in 16–19 April 2002 and the upstream peak discharges generated in the hydrologic model are used".

c) Damage analysis: for damage analysis we needed to have the land use and the inundation maps. The later was output of HEC-RAS model. Land use maps including residential buildings, restaurants, and agricultural areas were available from local municipality. Applying the inundation map on the land use maps the average depth of inundation and area of inundation for every land use were calculated. Then from the damage-elevation

curve percentile of damage to the land uses could be estimated. Finally, the damage cost to each land uses was calculated by the average economic value of one unit of that land use (available from a field survey and interviews with the local authorities and inhabitants).

P7L21-28: Damage Analysis: "In this section, with the help of GIS tool and the land use maps which were obtained from the local municipality, a simple analysis of damages to the buildings, their contents, and agricultural areas is carried out. In this step just five Sub-basins of Imamzadeh Davood, Rendan, Sangan, Sulaghan and Keshar are considered; because of lack of land use maps, low population, and low development in the other sub-basins. For this regard, applying the inundation map on the land use maps, the average depth of inundation and area of inundation for every land use are calculated. Then from the damage-elevation curves percentile of damage to the land uses can be estimated. Finally, the damage cost to each of land uses is calculated by the average economic value of one unit of that land use. It should be noted that for agricultural physical damages analysis in every sub-basin, two dominant products of cherry and apple were identified and based on the percentage of each of them, average crop number per unit area and value of each crop, the damage analysis was performed. Percentages of crops, number of them per unit area and their economic value as well as values of different assets in the flood plain are obtained by several field survey, interviews with the local authorities and local inhabitants, and engineering judgment.

P5L26: "Damage-Elevation curves that are prepared for different land uses of Kan River Basin are presented in Fig 1"

[Figure]

[Figure]

FIG 1 *DAMAGE-ELEVATION CURVES FOR DIFFERENT LAND USES OF A) BUILDING AND ITS CONTENTS, B) RESTAURANT'S CONTENT, AND C) AGRICULTURE IN KAN RIVER BASIN.*

**Comment #3:** In addition to the above, I have some difficulty understanding the motivation for this work. Clearly, atmospheric circulation patterns must be expected to impact extreme rain intensities. However, considering sufficiently long rainfall time series, these oscillations should not affect the probability distribution of extreme rainfall and thus also not our estimates of expected flood damages? Why then do we need to know exactly how much damages vary over time?

**Response:** We agree that high-intensity rainfalls are available in a long hydrologic time series and any high-intensity rainfall does not change the probability of its occurrence much. However, as the reviewer mentioned, rainfall events during El Nino are raised in intensity and amount. Therefore, even El-Nino event in a year does not expect to change the general probability distribution of extreme rainfall and the estimates of the flood damages, we could expect to have higher flood events and thus more estimates of flood damages in that year. This paper is looking to have an estimates of flood damages in such conditions in comparison with the

normal years. In other words, we want to calculate the peak values in the long time series of flood damages because we think that it is related to the El Nino.

**Methodological issues**

**Comment #4:** Linking rainfall series and SOI - The references I found describe the AMI as a univariate method and I could not find it in the reference provided by the authors. It is not clear how the joint probabilities are computed from the histograms and the equation for the "optimal number of categories"appears out of the blue". Most importantly, the authors did not provide any evidence that the method gives reasonable results (time series plots of SOI and rainfall indicating the identified lag, scatterplots of lag vs. AMI, cross correlation plots, or similar)

**Response**: We mentioned in the paper that average mutual information (AMI) is a measure of the "amount of information" obtained about one random variable, through the other random variable. Guiasu (1977) defined the mutual information of two random variables as a measure of the mutual dependence between two variables. Not limited to real-valued random variables and linear dependence like the correlation coefficient, mutual information is more general and determines how different the joint distribution of the pair (X,Y) is to the product of the marginal distributions of X and Y (Guiasu 1977).

Sometimes it is useful to express the mutual information of two random variables conditioned on a third called conditional mutual information. Therefore, mutual information among more than two random variables is also defined. Several generalizations of mutual information to more than two random variables have been proposed (McGill 1954; Hu Kuo Ting 1962). Therefore, AMI is not defined just as a univariate method; although, we have used it as univariate between SOI and precipitation in this paper.

When dealing with large sets of numbers, Sturge's rule (Sturges 1926) can be used to choose the number of categories. Sturge's rule is widely used in the statistical packages like excel for making histograms. According to Sturge's rule the data range should be split into $K$ equally spaced classes where:

$$K = 1 + 3.322 \log_{10}(n)$$

where $n$ is the number of data.

Reference

- Guiasu, S. 1977. Information Theory with Applications. McGraw-Hill, New York. ISBN 978-0-07-025109-0.
- Hu, K.T. 1962. On the Amount of Information. Theory Probab. Appl. 7: 439–447.
- McGill, W. 1954. Multivariate information transmission. Psychometrika. 19(1): 97–116. doi:10.1007/BF02289159
- Sturges, H. 1926. The choice of a class-interval. J. Amer. Statist. Assoc., 21, 65-66.

For verification of the result we can use the linear dependence method of correlation coefficient. Doing so, a scatter plot between monthly SOI and precipitation in Mehrabad synoptic station is prepared. Considering different lag times between SOI and precipitation, Fig 2 represents the correlation coefficient against monthly lag time. This figure is related to the precipitation in the Mehrabad station. Such a trend can be found in the other stations.

[Figure]

FIG 2. *CORRELATION COEFFICIENT BETWEEN PRECIPITATION AND MONTHLY LAG TIMES AGAINST DIFFERENT LAG TIME*

**Comment #5: Determining increased rainfall during El Nino**

(a) There is an obvious problem in using change factors derived for annual rainfall to extreme daily/hourly precipitation. No reasoning is provided for why this is done.

Response: In the present paper a risk base analysis has been performed. On this basis, results of Table 3 mean that for example if a 10-yr return period rainfall is happening while a strong El Nino condition is experiencing, it is expected by 90% probability that the damages cost will be less than 267% of that in the normal condition. Indeed, the rainfall and damage enhancements that are presented in the paper are the expected increase values of yearly rainfall and damages which have been calculated from a long time series of data. Therefore, the %-changes of damages represent the expected annual values for every return period or the values that are probable by the given certainty levels. Fig 3 illustrates the cumulative distributions that are fitted on the %-changes of rainfall in two different months: January and April. Noted that, according to the results El Nino in average increases the rainfall in April and decreases it in April. This figure shows that in 60% certainty the increased rainfall in Tehran is less than 36% in January and in April it follows not only an increased amount but a decreased value less than -28%. However, in the annual time scale which is reported in the paper for the same probability the rainfall increase is less than 8.2. On the other hand, in a year with dominant El

Nino, there are several La Nina events in which it is expected to have less flood damages. Therefore, in average, the annual expected increase in flood damages cannot be calculated by considering the effect of El Nino on a storm event happens in specific time of year and it requires to evaluate all the events during El Nino, La Nina, and neutral episodes. Certainly, it is better to employ a daily rainfall-runoff-damage analysis, but the absence of such data and information, applying the average annual rainfall increase (that is done in the paper), although is relatively simplistic, can provide acceptable results.

[Figure]

Fig 3. **Cumulative distribution function fitted on the increased percentiles of rainfall in a) January and b) April**

**Comment #6: Determining increased rainfall during El Nino**

(b) I believe Fig. 3 illustrates the %-changes of rainfall for all years that were identified as "El Nino" in step 1.

i. It is not clear to me why you would pick the 60 and 90% quantiles for the further analysis. The median value, as far as I can see, is 0. My conclusion would be that there is no evidence for an impact of El Nino on annual rainfall? Also the trends identified in Fig.2 look very questionable. Have you tested the significance of parameters?

**Response:**

**Picking the 60 and 90% quantiles**: Fig 4 (Fig 3 in the original paper) shows the cumulative distribution of precipitation increases (%) in the El Niño years compared to normal years. Toward a risk-based analysis to the flood damage resulting from El Niño, the probability of any precipitation and its increase percentile occurring in the long-term time series of the study area is important. For this reason Fig 4 has been developed and used to determine the requirements.

In this article it was required to estimate the average amount of damages and the maximum amount of damages that are expected per year. Therefore, a probability level representative of the maximum possible damage and a probability level representative of average damage caused by El Niño were selected. According to Fig 4, if we divide the precipitation increase into two classes of zero to 20% and 20 to 40%, the 60% probability level can represent the mean precipitation increase and the 90% probability level can be considered as the maximum precipitation increase.

[Figure]

**Fɪɢ 4**: **Gᴜᴍʙᴇʟ Cᴜᴍᴜʟᴀᴛɪᴠᴇ ᴅɪꜱᴛʀɪʙᴜᴛɪᴏɴ ꜰᴜɴᴄᴛɪᴏɴ ꜰɪᴛᴛᴇᴅ ᴏɴ ᴛʜᴇ ᴀɴɴᴜᴀʟ ɪɴᴄʀᴇᴀꜱᴇᴅ ᴘᴇʀᴄᴇɴᴛɪʟᴇꜱ ᴏꜰ ʀᴀɪɴꜰᴀʟʟ**

**Impact of El Nino on annual rainfall**: For evaluating the impact of El Nino on annual rainfall that is better to have a judgment based on the trend analysis instead of CDF plot. According to the yearly SOI time series, 9 El Nino events have been identified from 1951 to 2017. Out of these 9 years, 6 years have experienced increase in the precipitation and 3 years with decrease in the precipitation. Fig 5 provides the annual rainfall against the SOI values during El Nino years. It is clear that the annual rainfall increases as the SOI value decreases (stronger El Nino). In average one unit decrease in the SOI, will enhance 361 mm annual rainfall.

On the other hand, Fig 6 provides the annual rainfall against the SOI values for the total data (1951-2017). This figure illustrates a significant trend in the annual rainfall amount *vs* SOI. It is obvious that by decreasing the SOI value in which the El Nino event got stronger, the annual rainfall increases. Such a trend is also reported in many other papers.

[Figure]

**FIG 5. PERCENTILE OF RAINFALL CHANGES AGAINST THE SOI DURING EL-NINO YEARS IN MEHRABAD STATION**

[Figure]

**FIG 6. PERCENTILE OF RAINFALL CHANGES AGAINST THE SOI IN MEHRABAD STATION FOR THE TOTAL YEARS**

**Comment #7: Determining increased rainfall during El Nino**

(b) I believe Fig. 3 illustrates the %-changes of rainfall for all years that were identifed as "El Nino" in step 1.

ii. Are the annual rainfall and %-change values correlated? This would certainly impact the trend estimates in Fig. 2 and pose a challenge for distribution fitting in Fig. 3?

Response: Annual rainfall and %-change values of rainfall?!

[Figure]

**FIG 7- ANNUAL RAINFALL AGAINST THE CHANGE PERCENTILE**

About the cumulative distribution, we checked again the result of %-change values and the CDF to ensure the calculations. The data can be provided for further control if required.

**Comment #8. Hydrological and hydraulic modelling**

(a) The hydrological model was calibrated for a single event only, which is not good practice. Why is only one of the 3 stations used for calibration?

**Response:** For the flood of 15–18 April 2003 all hydrometric stations have been considered in the model evaluation, although the calibration has been performed on the basis of recorded discharges in station 3. For calibration the automatic calibration of HEC-HMS has been used in which a single discharge station is required for the purpose. For the three other floods, the discharges are not available because of the stations failure or incorrect operation. The results are shown in the following.

[Figure]

**FIG 8. THE OBSERVED AND SIMULATED FLOOD HYDROGRAPHS OF 15–18 APRIL 2003**

Also, In addition to the flood events which have been mentioned in the manuscript, two other events are also are considered for calibration and verification of the model. Floods of 2009 and 2011 with peak discharges of 34.4 m3/s and 54.1 m3/s. Noted that hydrographs of these two foods were not available in the Gage 1 and 2. These results are added to the paper.

Comparison between the simulated and observed flood hydrographs are shown in the following:

[Figure]

**FIG 9. OBSERVED AND SIMULATED FLOOD HYDROGRAPHS AT SULAGHAN STATION IN 15–17 APRIL 2009**

For the flood of 11-13 March 2011, a peak of 54.1 m$^3$/s has been estimated by Regional Water Company of Tehran.

[Figure]

**Fɪɢ 10. Oʙsᴇʀᴠᴇᴅ ᴀɴᴅ sɪᴍᴜʟᴀᴛᴇᴅ ғʟᴏᴏᴅ ʜʏᴅʀᴏɢʀᴀᴘʜs ᴀᴛ Sᴜʟᴀɢʜᴀɴ Sᴛᴀᴛɪᴏɴ ɪɴ 11–13 Mᴀʀᴄʜ 2011**

**Comment #9. Hydrological and hydraulic modelling**

(b) It is not clear for which areas the HEC-RAS simulation is performed (not highlighted in Fig.1), so I cannot evaluate whether the link between hydrological and hydraulic modelling setup makes sense.

**Response**: In this study, the hydrodynamic model of HEC-RAS was calibrated using the historical discharges and depths recorded in the hydrometric gauges for flood depth values. Flood hydrographs of the sub-basins with different return periods were simulated by the calibrated rainfall-runoff model (HEC-HMS) and the peak values were used as the boundary conditions for the HEC-RAS model. In this research, all the basin has been model in the HEC-HMS, then the HEC-RAS has been set-up for all the basin integrally. Schematics of the HEC-HMS and HEC-RAS models are provided in Fig 11 and Fig 12, respectively.

[Figure]

**FIG 11. SCHEMATICS OF THE HYDROLOGIC MODEL IN HEC-HMS**

[Figure]

**FIG 12. SCHEMATICS OF THE HYDRADYNAMIC MODEL IN HEC-RAS**

**Comment #10.** Damage calculation – It is not clear how the damage calculation was performed. The depth-damage functions are not provided in the paper. In the results section, the authors

mention the computation of an "average inundation depth" per landuse class. This seems like a questionable approach, but it is simply not clear what was done here.

**Response**: Damage analysis part is rephrased and extended to clarify the methodology of damage estimation. Also damage-elevation curves are added to the paper.

P6L1: A comprehensive analysis of physical damages due to flooding requires many information including accurate updated land use map, area and age of buildings, type of the structure, number of floors, exact areas of different agricultural crop in the flood-prone area, crop number per unit area, value of crops, value of buildings (residential and non-residential) and their contents, number of residential, administrative, and commercial buildings in flood prone areas, the area and elevation of buildings, their locations, and spatial distribution of flood depth values in the inundated areas for different return periods. In this paper a simplistic approach is used for this regard. For the building damage analysis, separating residential and commercial ones, the total area of inundated buildings, average inundation depth, and the average economic value of the building and their contents for every buildings type are used. For agricultural damage analysis, considering the dominant crops of cherry and apple, the area of inundation, average inundation depth, crop density, and average price of one single crop the flood damage costs are evaluated.

P7L21-28: Damage Analysis: "In this section, with the help of GIS tool and the land use maps which were obtained from the local municipality, a simple analysis of damages to the buildings and their contents, and the agriculture is carried out. In this step just five Sub-basins of Imamzadeh Davood, Rendan, Sangan, Sulaghan and Keshar are considered; because of lack of land use maps, low population, and low development in the other sub-basins. For this regard, applying the inundation map on the land use maps, the average depth of inundation and area of inundation for every land use category are calculated. Then from the damage-elevation curves, percentile of damage to the land uses can be estimated. Finally, the damage cost to each of land uses is calculated by the average economic value of one unit of that land use. It should be noted that for agricultural damage analysis in every sub-basin, two dominant products of cherry and apple were identified and based on the percentage of each of them, average crop number per unit area and value of each crop, the damage analysis was performed. Percentages of crops, number of them per unit area and their economic value as well as values of different assets in the flood plain are obtained by several field survey, interviews with the local authorities and local inhabitants, and engineering judgment.

P5L26: "Damage-Elevation curves that are prepared for different land uses of Kan River Basin are presented in Fig 13"

[Figure]

FIG 13. *DAMAGE-ELEVATION CURVES FOR DIFFERENT LAND USES OF A) BUILDING AND ITS CONTENTS, B) RESTAURANT'S CONTENT, AND C) AGRICULTURE IN KAN RIVER BASIN.*

**Minor comments**

**Comment #11:** P1L27: How do you define a flood event? Is it correctly understood that Tehran experienced flooding 12 times in 1951 and 54 times in 1991?

**Response**: Tehran metropolis is the Iran's capital located below Alborz Mountains where several steep rivers flow through the city from the north to the south, named Kan River (the biggest), Farahzad River, Darband River, Darakeh River, Dar-Abad River and etc. Although they are not very big rivers, flooding is a major problem of them because of their steep basin, potential of snow melt, mountainous valleys that has forced people to live near the river beds and low flood management practices. These rivers have provided some populated rural areas and attracted many tourists for spending their times in the vacations. Todays, the population areas near the rivers have been expanding and the economic activities in the flood plains is being grown up. All of these factors have caused the flooding being increased. Due to flooding of a river/rivers, some parts of the city (areas close to the flooding river and along the river) may experience flooding. Therefore, in the mentioned sentence we do not mean that all the Tehran regions experience flooding in a flood event. To prevent misleading the readers, we rephrased the sentence as follow:

> "According to the available reports, the number of flooding events that happened in any parts of Tehran over four decades from 12 cases in 1951 had grown up to 54 cases in 1991"

**Comment #12:** P2L1-23: This part of the introduction cites a lot of studies that measured impacts of atmospheric circulation patterns. However, most of these refer to completely different parts of the world, so I had difficulty seeing the relevance.

**Response**: Many studies have shown the effect of ENSO on the climate variability of Iran. Nazemosadat and Ghasemi (2004) indicated that El Niño is associated with wet periods over most regions of Iran during autumn and winter while the risk of droughts is high during La Niña. Their study revealed that El Niño has the least influence over the southeastern and northwestern regions of the country during winter. Alizadeh-Choobari et al. (2018) indicated that the ENSO cycle contributes to the interannual climate variability over Iran. According to their results, about 26% of the variance in annual precipitation over Iran is related to the El Niño. Based on their achievements, In spite of the seasonality of the ENSO signal and its interevent variability,Iran is anomalously wet during the EP El Niño and dry during La Niña and the impacts of La Niña and the EP El Niño are generally stronger over the warm and arid regions of Iran.

Although, the effect of ENSO on the precipitation has been frequently studied in Iran (Nazemosadat and Ghasemi 2004; Saghafian et al. 2017; Alizadeh-Choobari et al. 2018; Hooshyaripor et al. 2018), there are few studies about ENSO influence on the socioeconomic impacts of floods even around the world (Ward et al. 2014). The main reason for the limited research on the economic impacts of climate and hydrologic variability is said to be the lack of economic data on flood damages (Changnon 2003). Analyzing the National Flood Insurance Program daily claims and losses and Multivariate ENSO Index (MEI), Corringham

and Cayan (2019) quantified insured flood losses across the western United States from 1978 to 2017. They showed that in coastal Southern California and across the Southwest of the United States, El Niño has had a strong effect in producing more frequent and higher magnitudes of insured losses, while in the Pacific Northwest, the opposite pattern with weaker and less spatially coherent has been reported. Changnon (2003) revealed that the strong El Niño events of 1982/83 and 1997/98 have caused significant flood damages over $2.8 billion in Southern California. Null (2014) demonstrated that from 1949 until 1997 out of the six seasons that flood damages costs exceeded $1 billion in California three cases had been El Niño years; one very strong (1982), one moderate (1994) and one weak (1968). Ward et al. (2014) showed that ENSO exerts strong and widespread influences on both flood hazard and risk. They assessed ENSO's influence in terms of affected population, gross domestic product and economic damages on the flood risk at the global scale and showed that climate variability, especially from ENSO, should be incorporated into disaster-risk analyses and policies. They revealed that, if the frequency and/or magnitude of ENSO events were to change in the future due to climate change, change in flood-risk variations across almost half of the world's terrestrial regions is happened. Ward et al. (2016) provided a global modelling exercise to examin the relationships between flood duration and frequency and ENSO. They indicated that the duration of flooding compared to flood frequency is more sensitive to ENSO.

Alizadeh-Choobari O., Adibi, P. and Irannejad, P. 2018. Impact of the El Niño–Southern Oscillation on the climate of Iran using ERA-Interim data. Climate Dynamics. 51(7-8): 2897–2911.

Changnon, S. 2003. Measures of economic impacts of weather extremes. Bull. Amer. Meteor. Soc., 84, 1231–1235, https://doi.org/10.1175/BAMS-84-9-1231.

Corringham, T.W. and Cayan, D.R. 2019. The Effect of El Niño on Flood Damages in the Western United States, Weather, Climate, and Society, 11(3), 489-504. https://doi.org/10.1175/WCAS-D-18-0071.1.

Nazemosadat, M.J., Ghasemi, A.R. 2004. Quantifying the ENSO-related shifts in the intensity and probability of drought and wet periods in Iran. J Clim 17:4005–4018

Null, J. 2014. El Niño and La Niña: Their Relationship to California Flood Damage, Golden Gate Weather Services, August 2014.

Ward, P.J., Jongman, B., Kummu, M., Dettinger, M.D., Sperna Weiland, F.C. and Winsemius, H.C. 2014. Strong influence of El Niño Southern Oscillation on flood risk around the world, Proceeding of National Academy of Sciences of America (PNAS), 111(44), 15659-15664.

Ward P.J. Kummu M., Lall U. 2016. Flood frequencies and durations and their response to El Niño Southern Oscillation: Global analysis, J Hydrology, Volume 539, August 2016, Pages 358-378.

**Comment #13:** P3L10: typo 78.23M cbm/s?

**Response**: Thank you. It is annual inflow that was measured in Sulaghan station equal to 78.23 Mm$^3$/yr.

**Comment #14:** P4L20-25: What is a natural uniform rainfall?

**Response**: "naturally", we mean the rainfall that has been happened historically in the basin. "Uniform", we mean that the rainfall has the same spatial intensity and duration. This term has been changed in the revised paper as:

Scenario I (normal condition): In the first scenario no El-Niño event is considered. It is assumed that the basin receives a rainfall with the given intensities (T=10, 25, and 50) in *Tc* min duration.

---

## Author Comment (AC3) · 13 Dec 2019

**Response Letter to Reviewers Comments on nhess-2019-166**

Many thanks for the quick response from the editor and the reviewers. The manuscript has been improved substantially based on the constructive comments of the reviewers.

(**The highlighted parts are added to the revised paper**)

**Response to Comments of Reviewer #3**

**General Comment:** The manuscript described an interesting idea on looking at the flood damage caused by ENSO. A case study in Kan River Basin, Iran was presented. The manuscript is well organized and easy to follow. However, I think the current results are not convincing enough, because huge uncertainties from the 6-step were ignored. Extrapolation without acknowledging uncertainties can mislead the result. Thus, I recommend return the manuscript to authors for major revision.

**Response:** Many thanks for your comments. The manuscript has been improved substantially based on your constructive comments.

**Major comments:**

**Comment #1:** In methodology section, authors presented a 6-step method to investigate the flood damage from ENSO. In each of those steps, there are uncertainties. In particular, the paper steps I, II, III and VI are based on probabilistic models, in which uncertainties cannot be avoid. The main issue is that when putting those 6 steps in series the uncertainty can be exploded. This makes the result meaningless when having huge uncertainty. When estimating the relationship between rainfall variation and SOI, large uncertainty should exist on the slope as shown in Figure 2. The results derived from Eq 3 should also have a large uncertainty. When bringing this uncertainty into next steps, the final results may be very different from what has been found right now.

**Response**: Thank you for your valuable comments over the methodology. We agree that there are uncertainties in every step. However, most of the uncertainties can be assessed quantitatively or constrained by the observations. These uncertainties have been quantitatively assessed in the paper, e.g. considering different probability for %-increase in the rainfall. In the revision, we have summarized the uncertainties in every step and provided brief discussions on the limitations:

Step I: noted that there is uncertainty in the optimal number of categories that may influence the lag time between the precipitation in the basin and SOI.

Step II: Rainfall increases in the El Nino years are different and considering an average value for the flood damage analysis cannot provide an acceptable result. It is better to consider the

uncertainties in a specific way. The employed probabilistic method for considering the rainfall increase percentiles can cover some of these uncertainties.

Step III: This is a limitation in our methodology in which the increase percentile for the rainfall for every return period has been considered the same.

Step VI: In this section, with the help of GIS tool and the land use maps which were obtained from the local municipality, a simple analysis of damages to the buildings and their contents, and the agriculture is carried out. In this step just five Sub-basins of Imamzadeh Davood, Rendan, Sangan, Sulaghan and Keshar are considered; because of lack of land use maps, low population, and low development in the other sub-basins. For this regard, applying the inundation map on the land use maps, the average depth of inundation and area of inundation for every land use category are calculated. Then from the damage-elevation curves, percentile of damage to the land uses can be estimated. Finally, the damage cost to each of land uses is calculated by the average economic value of one unit of that land use. It should be noted that for agricultural damage analysis in every sub-basin, two dominant products of cherry and apple were identified and based on the percentage of each of them, average crop number per unit area and value of each crop, the damage analysis was performed. Percentages of crops, number of them per unit area and their economic value as well as values of different assets in the flood plain are obtained by several field survey, interviews with the local authorities and local inhabitants, and engineering judgment.

**Minor comments:**

**Comment #2:** There are some constant values in equation 2 and 4. Authors need to justify these numbers.

**Response:** We have now justified these numbers as follows:

Equation 4: The constant value 0.2 in Eq. (4) is selected based on SCS recommendation. SCS has proposed the initial losses can be estimated as $I_a$=0.2$S$ (Ponce and Hawkins 1996). We did not have any estimate for this losses therefore, the SCS recommendation was taken for Kan River Basin.

$$P_e = \frac{(P - I_a)^2}{P + 0.8S}$$

Substitution of $I_a$=0.2$S$ in the above equation

$$P_e = \frac{(P - 0.2S)^2}{P + 0.8S} \qquad \text{Eq. (4)}$$

where P is rainfall; and S is storage potential. Moreover, Clark instantaneous unit hydrograph method is applied to transform the effective rainfall into runoff (Q).

Equation 2: When dealing with large sets of numbers, Sturge's rule (Sturges 1926) can be used to choose the number of categories. Sturge's rule is widely used in the statistical packages like excel for making histograms. According to Sturge's rule the data range should be split into $K$ equally spaced classes where:

$$K = 1 + \log_2(n)$$

where $n$ is the number of data. Changing the logarithm to the base of 10, we have:

$$K = 1 + 3.322\log_{10}(n) \qquad \text{Eq. (2)}$$

Reference:

- Ponce, V. and Hawkins, R. 1996. Runoff Curve Number Has It Reached Maturity Journal of Hydrologic Engineering, 1, 11-19.
- Sturges, H. 1926. The choice of a class-interval. J. Amer. Statist. Assoc., 21, 65-66.

---

## Author Response (AR2)

**Response Letter to Reviewers Comments on NHESS-2019-166#R2**

**Dear Prof. Merz**

Many thanks for the constructive comments and suggestions from you and the reviewers. We carefully considered all issues mentioned in the reviewer's comments, and we outlined every change point by point, as highlighted in the reversion. We believe that the reviewer's comments and suggestions have helped us to improve the quality and readability of the paper. The point-by-point responses are provided below.

(**The highlighted parts are added to the revised paper**)

**Responses to reviewer #1**

1- **Section 3. The presentation of the six steps need some improvements. Maybe it is better to have a purpose of each step, then followed by a description of how this purpose is achieved (methods, equations, etc.). For example, Step I, the purpose is to estimate the lag time. However, I do not understand how AMI method presented below can be used to estimate the lag time. Step II, how to estimate the influence of ENSO is not described, by regression or some other methods? Now there is too much information mixed in this part, which made the it hard to read and follow**

**Reply:** It is corrected according to the reviewer comment

Step I: **(P5L3-8)** *As the effect of ENSO takes time to be experienced in far geographic locations, the lag time between the ENSO occurrence and the related influences in Kan River Basin was firstly calculated. This lag time can be revealed by comparison the variations of SOI and local precipitation time series. The monthly rainfall at the nearby synoptic stations of Mehrabad (1951-2017), Shemiran (1988-2017), Tehran-Geophysics (1992-2017) and Chitgar (1997-2017) (See Figure 1) and monthly SOI values are used. A statistical method, the average mutual information (AMI), is used to determine the time delay.*

Step II: **(P6L4-6)** *Secondly, the influence of El-Niño on the precipitation amount in Kan River Basin is quantified. The influence is estimated using a statistical*

*method by calculating the expected value of the changes of precipitation amount in the El Nino episodes compared to those in the neutral periods.*

Step III: **(P6L23-25)** *Thirdly, several design storms are generated to be applied in a rainfall-runoff model. The rainfall storms are synthesized based upon the average precipitation change during El-Niño events. The designed storms are used for assessing the flood damages in a certain return period.*

Step IV: **(P7L14-16)** *Fourthly,* the HEC-HMS hydrologic model is used to simulate the rainfall-runoff process. *The hydrologic model is run for every scenario and every return period; then the peak discharges are used in the next step to estimate the flooding depths. In the hydrologic model,* the SCS method is used to calculate the effective rainfall.

Step V: **(P8L4-5)** *Fifthly, based on the obtained flood depth, the flooding areas are determined for designed storms in the El Nino and neutral periods.*

Step VI: **(P8L9-10)** *Finally, flood damage is assessed for all 9 runs of the model. These damages can be compared to each other in order to determine the role of El-Nino on the flood damages.*

2- **Page 4, lines 11, error on the reference.**
   **Reply:** corrected

3- **There are two Table 1 in the manuscript.**

   **Reply:** corrected

**Responses to reviewer #2**

**1- Fig. 4 in the paper clearly illustrates that the median % change of precipitation during El Nino years is equal to 0.**

Reply: Figure 3 of the paper shows that the occurrence of El-Nino increases the precipitation, and this increased value is significant (see Table 1). However, we confirm the median in Figure 4 is close to zero (4%) and the mode value (for which the probability density function is maximum) is 10.1% **(P9L20-21).**

[Figure]

**Figure 1. Probability density function (PDF) of %-increased rainfall values under the effect of El Nino (SOI<-0.8)**

[Figure]

**Figure 2. Cumulative distribution function (CDF) of %-increased rainfall values under the effect of El Nino (SOI<-0.8)**

**(P9L21-23)** In fact, the reason for that the median is close to zero is due to the selection criterion of 9 events as the El Nino events out of total events; (SOI less than -0.8 according to Australia Bureau of Meteorology). If this criterion is set as SOI<-1.0 (according to the Western Regional Climate Center, USA), the number of El Nino events will decrease to 6. Using frequency analysis approach, the probability distribution of data is as Figure 3:

[Figure]

**Figure 3. CDF of %-increased rainfall values under the effect of El Nino (SOI<-1.0)**

**(P9L23-26)** As can be seen, in this condition the median increases to 12.2; because two El Nino events with negative %-increased precipitation which had a great impact on the results, were eliminated. Therefore, there will be just one El Nino event (in which SOI = -1.018) with negative %-increased precipitation (-33%) in the data. It is clear that if the criterion for determining the occurrence of El Nino be changed as SOI< -1.02, the only negative value would be omitted and in this case, there would be 5 El Nino events in the analysis. Therefore, the probabilistic distribution on the %-increased values changes as follow:

[Figure]

**Figure 4. CDF of %-increased rainfall values under the effect of El Nino (SOI<-1.018)**

It is clear that in the new situation, the median will increase significantly to about 20 **(P9L25).**

The purpose of the above explanation is to state that in the analysis used in this paper, the criterion for distinguishing the El Nino condition is the criterion for distinguishing the El-Niño condition is an effective assumption in this paper and it affects the results significantly. Therefore, the results of Figure 4 (in the paper) should not be evaluated as the insignificant effect of El Nino on the Kan River Basin precipitation **(P9L25-27)**. Certainly, to assess the impact of El Nino on the amount of annual rainfall, Figure 3 (in the paper) can be used in

which the trend of annual rainfall against the SOI variation (without omitting any data) has been shown.

**2- Taking the 60 and 90% quantiles of this distribution and declaring these as representative for flood risk changes due to El Nino is simply not a valid approach, because it completely ignores the lower tail of the distribution where rainfall during El Nino years is actually decreased. I included some possible options for fixing this below, but the author's may have different ideas. In any case the risk analysis probably needs to be redone, so a major revision will be required.**

**Reply**: We agree with the opinion of the reviewer. To have a comprehensive study on the risk of flood due to El Nino, it is required to consider the whole range of probabilities: in both magnitude of hazard (flood) and magnitude of El Nino **(P12L16-18)**. But at first, as explained previously, Figure 4 is not enough to show the impact of El Nino on the precipitation and it is required to considered all the time series of SOI for such judgment. In the current condition, taking all range of %-increased precipitation probabilities consisting the lower tail of distribution where rainfall during El Nino years is actually decreased and then numerically integrate over the distribution will be close to the median of the distribution **(P12L21-24)** which is less than those calculated for 60 and 90 percentiles. Secondly, one of the purposes of this article was to show the importance of small floods (floods with a low return period) in flood management plans, and for this reason we have compared 5-yr and 10-yr floods with larger flood of 50-yr return periods (These values are selected in accordance with the paper's objective to show the importance of small floods (floods with a low return period) in flood management plans compared to the high return period floods) **(P6L28-29)**. Therefore, three floods with specific return periods (low to high return periods) are considered to cover the range of probability for flooding. This means that in terms of flood risk and probability of flood occurrence, the concern of the reviewer about the magnitude of hazard has been addressed. Therefore, by comparing the 5-yr and 10-yr flood with floods of larger return periods, the importance of El Nino on the floods with the low return period and the resultant damages has been shown and emphasized in the "abstract" and "results and discussion" sections.

However, about the full coverage of the probabilistic distribution of %-increased precipitations due to the El Nino (Figure 4), it should be noted that the probability levels of

60% and 90% are considered in accordance with the purpose of the paper. The objective of the paper as has been mentioned in the end of introduction **(P3L25-26)** is to show the amount of flood damage **that can be added (possible increase in the damages)** due to the El Nino. These percentiles are not representative for flood risk changes due to El Nino but are the selective high probability levels represent for the highest influence of El Nino with low probability levels of occurrence. This objective was mentioned in the paper as **(P7L9-13)**:

> "*The reason ==behind the choice of== 60% and 90% probability levels is to estimate the average amount of damages and the maximum amount of damages that are expected per year due to moderate to strong El-Niño events. Therefore, a probability level representative of the maximum possible damage and a probability level representative of average damage caused by El Niño were selected.* "

Indeed, the fact that El Nino increases the precipitation amount has been proven in the previous section of the paper (Figure 3 and related explanations), and there is no doubt that there is a significant direct relationship between precipitation and the magnitude of El Nino (see Table 1). In this part of the paper, we seek to show the **possible increase** in flood damages due to El Nino, not the possible decrease value and not the expected value. This objective was highlighted in the abstract as **(P1L16-18)**:

> "*To determine the flood damage costs, the ==annual== precipitation enhancement during El-Niño condition was ==firstly== estimated using a probabilistic approach and the inundation area was then determined under ==high probability levels of increased rainfall due to El Nino== for 5-, 10- and 50-year return period floods.*"

Also in the introduction section, it has been mentioned that **(P3L25-26):**

> "*The question ==addressed== in this research is that, given the increasing impact of rainfall due to El-Niño, how much losses/damages ==are== expected to be added in a specified study area.*"

In this regard, we selected two percentiles in the upper tail of probabilistic distribution: 1) increase in the probability level of 60%, and 2) increase in the probability level of 90%; then we compared the results with those in the normal condition. Doing so, three return periods in three scenarios of El Nino event were defined and a total of 9 model runs were performed.

These results provide decision makers with essential information on flood risk and highlight the importance to take in to account the probable effect of El Nino in flood risk management (P13L1-4).

Also, in some other studies of flood risk analysis, depending on the purpose of the study some parts of the probability distribution may have been neglected. For example:

To evaluate the risk of flood using a fuzzy approach, Nandalal and Ratnayake (2011) just selected rainfalls of a 100-year return period. They did not consider the lower and higher tails of the probability distribution. Grabs (2015) evaluated the flood risk reduction measures in the Elbe River for 100-yr flood. He took two 100-yr return period floods in 2002 and 2013 and did not analyzed the other floods with lower return periods. Noted that, the flood event of 2002 in the entire Elbe River Basin has become a reference in Europe for extreme flood events so in this paper the extreme floods were analyzed. Van Dau et al. (2017) evaluated only 25-yr return period precipitation to quantify flood damage under potential climate change impacts in central Vietnam neglecting the other floods with lower or higher frequencies. To consider the effect of climate change on flood risks, Van Dau et al. (2017) considered the HadGEM3–RA Regional Climate Model (RCM) under two Representative Concentration Pathways (RCP) 4.5 and 8.5 climate change scenarios neglecting RCP 2.6 a representative for the lower tail of climate change scenarios.

Apart from this, as probability distribution of Figure 4 is not a full representative for El Nino effect on the precipitation, to make the point of view of the respected reviewer (regarding the study of the entire range of probability distribution), it is better to consider the whole range of ENSO without excluding some smaller events of El Nino (may be -0.8<SOI<0) and/or events of La Nina. In this case, it is necessary to use a different methodology in another independent study. The following figure shows the probabilistic distribution of SOI values. Figure 5 shows the PDF (here normal distribution) and Figure 6 shows the CDF. As shown, changes in the SOI index follow a normal distribution. The whole probability range can now be considered to examine the impact of ENSO on flood damage.

[Figure]

**Figure 5. PDF of entire ranges of SOI values**

[Figure]

**Figure 6. CDF of entire ranges of SOI values**

3- **The above assumes that it is a valid approach to apply annual precipitation change factors directly to extreme rainfall on short time scales. This is another major**

**limitation of the paper which needs to be stated clearly in the discussion and conclusions. The author's quite nicely illustrate this in their reply to reviewer 1 (comment 1) in that average change of monthly precipitation in El Nino years is +36% in January, but -28% in April. The former is likely to be linked to snowfall, while the latter actually suggests a decrease of rainfall during the flooding season.**

**Reply:** this limitation is explained in the paper as **(P6L15-20):**

*PC values then will be used to construct synthesized rainfall storms for simulation of the El-Niño influence. It is a major limitation of this research that the annual change factor is applied in the extreme rainfalls of short time scales. Certainly, it was better to consider the monthly or seasonal change factor or calculation of change factor on the basis of recorded storms the applying it in a continues hydrologic model to have a more accurate prediction of El-Niño effect on the flood damages then calculation of the annual damages over years, but because of data limitation the analyses performed for the annual data.*

**Major comments:**

4- **Quantification of rainfall changes - I could imagine 2 ways of doing this:**

- **Compute flood risk for 10 or so quantiles of the distribution in Figure 4 (considering positive AND negative changes) and then numerically integrate over the distribution.**

- **Compute flood risk for each of the 9 El Nino years and compare average against non El-Nino years.**

**Reply:** please see the reply to comment Number 2.

5- **Merge Fig. 5 and/or 6 from the author's response into Fig. 3 in the paper. These are much more illustrative, while the data in Fig.3 currently may or may not support the assumption of a trend.**

**Reply:** Fig 3 was modified and Fig6 (in the previous author responses) was added.

Also the following text was added to the paper (**P9L4-10**):

*"In Figure 3, the annual rainfall of stations is plotted against the SOI index. It is obvious that with decreasing SOI index, annual rainfall increases in the study area and vice versa. In the period of 1951 to 2017, a total of 9 El-Niño*

*(SOI<-0.8) and 7 La-Niña (SOI>+0.8) events have been occurred. ==Out of them, 6 years have experienced increase in the precipitation and 3 years with decrease in the precipitation.== The largest event for El-Niño dates back to 1983 and 1987 with respectively 334 mm and 252 mm recorded rainfall in ==Mehrabad station. Furthermore, based on the trendlines, in average one unit decrease in the SOI, will enhance 22.5 mm annual rainfall in Mehrabad station.== For further analyses, Mehrabad station was chosen because it has more data than the other stations.''*

[Figure]

**Figure 3- Annual rainfall against SOI index in the station of a) Mehrabad, b) Shemiran, c) Chitgar and 4) Tehran Geophysics**

**Minor comments:**

6- **p.4 l.13: all your result figures suggest annual rain depths around 250-300mm, so I'm surprised to see 640mm here?**

**Reply:** Kan River Basin is part of a larger basin. The larger basin includes Mehrabad, Shemiran, Chitgar and Tehran Geo-physics rain stations in lower levels than the Kan River Basin. The basin is a small mountainous basin located at upstream. Therefore, the average annual precipitation of Kan River Basin is more than those presented in Fig.3.

**7- Eq.2: it is still unclear how you compute the probabilities. Please specify, possibly in an appendix**

**Reply:** The Average Mutual Information (AMI) measures how much one random variable tells us about another. In the context of time series analysis, AMI helps to quantify the amount of knowledge gained about the value of *x(t+λ)* when observing *x(t)*. Since mutual information can be computed for a times series and a time-shifted version of the same time series, this is called the auto mutual information. However, it can be calculated for two different time series as average mutual information.

To measure the AMI of a time series, a histogram of the data using bins is created. Let *Pi* the probability that the signal has a value inside the i*th* bin, and let *Pij(λ)* be the probability that *x(t)* is in bin *i* and *x(t+λ)* is in bin *j*. Note that only the joint probability *Pij(λ)* depends on λ, and that the AMI function also depends on how the histograms are constructed, i.e., the width and position of the bins. Then, AMI for time delay λ is defined as

$$AMI(\lambda) = sum(\ Pij\ log(\ Pij\ /\ (Pi*Pj)\ )\ )$$

Depending on the base of the logarithm used to define AMI, the AMI is measured in bits (base 2, also called shannons), nats (base e) or bans (base 10, also called hartleys). In this paper shannons type was used and probabilities $P_A$ and $P_B$ were calculated using empirical frequency analysis in which the relative frequency histograms for both time series, SOI and precipitation were determined. The values of AMI for different arbitrary lag-times (1 to 12 months) between SOI and precipitation were calculated. The higher AMI value, the more dependency between two time series. Therefore, that lag-time corresponding to the highest AMI value was selected as the lag-time between the time series **(P5L19-23)**.

**8- p.8 l.25: This paragraph describes methodology which should be merged into the description of assessment steps (probably step 3)**

**Reply:** the paragraph was revised and those parts related to the methodology section were moved to the methodology section step II.

"Methodology" **(P6L30-P7L2)**

> To determine the rainfall intensity in every scenario, PC values are employed and using an appropriate analytic probability distribution, the rainfall increase in different confident levels are determined. Here two probability levels, 60% and 90%, are considered for every rainfall return period. Accordingly, 9 different model runs were evaluated in the following scenarios:

"Results and discussion" **(P9L17-20)**

> Then, using Eq. (4), PC and $\Delta P$ can be calculated. According to the results, for 9 years with El-Niño condition, PC ranges from -60.34% to 42.8% while the latter is related to the year 1983 in which 334 mm rainfall was recorded. On the basis of Kolmogorov-Smirnov goodness of fit test with 99% certainty, Gumbel distribution well fits on these percentiles (Figure 4).

9- **the motiviation of the paper is still not quite clear. I would say we are trying to identify the potential variability due to El Nino to be able to separate from other (e.g., climate) effects?**

**Reply:** the objective of the paper has been mentioned in Abstract and in the end of the introduction as:

Abstracy **(P1L14-18)**

> This study aims at determining the effect of the most emblematic teleconnection, El-Niño, on the expected damages of floods with low return periods in Kan River basin, Iran. To determine the flood damage costs, the annual precipitation enhancement during El-Niño condition was firstly estimated using a probabilistic approach and the inundation area was then determined under high probability levels of increased rainfall due to El Nino for 5-, 10- and 50-year return period floods.

Introduction **(P3L22-23)**

> The question addressed in this research is that, given the increasing impact of rainfall due to El-Niño, how much losses/damages are expected to be added in a specified study area.

**10- the paper does require language revisions if accepted**

**Reply****:** Done.

**References**

Grabs, W. (2015). Benchmarking flood risk reduction in the Elbe River. Journal of Flood Risk Management, 9(4), 335–342. doi:10.1111/jfr3.12217.

Nandalal, H. K., & Ratnayake, U. R. (2011). Flood risk analysis using fuzzy models. Journal of Flood Risk Management, 4(2), 128–139. doi:10.1111/j.1753-318x.2011.01097.x

Van Dau, Q., Kuntiyawichai, K., & Plermkamon, V. (2017). Quantification of Flood Damage under Potential Climate Change Impacts in Central Vietnam. Irrigation and Drainage, 66(5), 842–853. doi:10.1002/ird.2160

Western Regional Climate Center, USA; https://wrcc.dri.edu/

---

## Author Response (AR3)

**Dear Prof. Merz**

Many thanks for the constructive comments and suggestions from you and the reviewers. We carefully considered all issues mentioned in the reviewer's comments, and we outlined every change point by point, as highlighted in the reversion. We believe that the reviewer's comments and suggestions have helped us to improve the quality and readability of the paper. The point-by-point responses are provided below.

(**The highlighted parts are added to the revised paper**)

**Reviewer #2**

I'm sorry to send this back one more time, but the statistical analysis is still not valid. It becomes very clear when we look at Fig. 3. The current approach in the paper is the following:

1. Take all rainfall values with SOI below -1

2. Compute % change in annual rainfall for each of these years compared to mean annual rainfall for the entire series

3. Fit a distribution to these values and use the 60 and 90% quantiles for evaluating the impact of El Nino.

Step 3 implies that the authors find the "60 and 90% biggest rainfalls" during El Nino years and evaluate how much bigger these are than the average annual rainfall. However, these quantiles are simply an expression for random variation in the data (we can see similar levels of variation in non-El-Nino years. In fact, the 3 years with highest annual precipitation are outside the El Nino season).

The correct question to ask would be:

How much bigger is the mean annual precipitation in El-Nino years compared to the mean annual precipitation in non-El-Nino years?

If have done a quick bootstrap and permutation test on the Mehrabad dataset (R script inserted below). The answer is that when we consider a SOI threshold of -0.8, there is no difference between the mean annual precipitation in El-Nino and non-El-Nino years. Considering a threshold of -1, there is a mean difference in the order of 11% (which is in line with the median

difference mentioned in the paper). The statistical evidence for the difference in mean annual precipitation is weak - in the permutation test it is significant at a confidence level of 63%.

I suggest the following:

-Remove any mentioning of confidence levels in the paper (abstract, page 7, discussion, conclusions)

-Consider the 12% median change in annual precipitation as an expression for the change inflicted by El Nino and consider only these results.

In addition to the above, please mention clearly in the abstract and the outlook that

-Annual change factors cannot necessarily be transfered to extreme values.

-In fact, your own data suggest that when considering monthly rainfall the effect of El Nino might negative when considering periods where extreme rainfall occurs. These are badly understood effects that require more research.

Reply: Thank you for your precision and your help to improve the paper. All of your suggestion has been considered in the revised paper. Many parts of the paper have been changed and the added parts have been highlighted by blue color. In the revised paper the following practices have been done according to your suggestion:

1- **The confidence levels including Figure 4 (CDF curve) and distributions were removed and any explanation in this regard was omitted.**

2- **The topic of the paper was revised according to the revision as**

"Annual flood damage influenced by El-Niño in Kan River Basin, Iran"

3- **The median change in annual precipitation as an expression for the change inflicted by El Nino was calculated equals to 12.2% and the paper based upon this percentile was revised. Abstract, Methodology, Results and Conclusion parts were revised totally considering only this median value.**

3-1- **Abstract (P1L16-24)**

[revised manuscript text omitted]
 the abstract and Conclusion the limitation of our study as you mentioned were clearly mentioned:**

P1L16-18

To determine the flood damage costs, the median of annual precipitation changes during El-Niño condition was firstly estimated, although the annual precipitation change factor cannot necessarily be transferred to extreme values.

**P10L24-26**

Noticed, the annual change factor cannot necessarily be transferred to extreme values. While, considering monthly rainfalls the effect of El Niño might be negative in some periods when extreme rainfall occurs.

---

## Author Response (AR4)

**Response Letter to Reviewers Comments on NHESS-2019-166#R4**

**Dear Prof. Merz**

Many thanks for the quick review and constructive comments on the manuscript "Annual flood damage influenced by El-Niño in Kan River Basin, Iran" from you and the reviewers. We carefully checked the details and language of the manuscript and improved its context with the help of a native English speaker. The authors would like to thank you again for your time and consideration.

Regards